# Histone demethylase KDM2A recruits HCFC1 and E2F1 to orchestrate male germ cell meiotic entry and progression

Shenglei Feng[1,2,7], Yiqian Gui[1,7], Shi Yin [iD][3,7], Xinxin Xiong[1], Kuan Liu[1], Jinmei Li[1], Juan Dong[4], Xixiang Ma[2], Shunchang Zhou[2], Bingqian Zhang[1], Shiyu Yang[1], Fengli Wang[1], Xiaoli Wang[1], Xiaohua Jiang [iD][5✉] & Shuiqiao Yuan [iD][1,2,6✉]

## Abstract

**In mammals, the transition from mitosis to meiosis facilitates the successful production of gametes. However, the regulatory mechanisms that control meiotic initiation remain unclear, particularly in the context of complex histone modifications. Herein, we show that KDM2A, acting as a lysine demethylase targeting H3K36me3 in male germ cells, plays an essential role in modulating meiotic entry and progression. Conditional deletion of *Kdm2a* in mouse pre-meiotic germ cells results in complete male sterility, with spermatogenesis ultimately arrested at the zygotene stage of meiosis. KDM2A deficiency disrupts H3K36me2/3 deposition in c-KIT[+] germ cells, characterized by a reduction in H3K36me2 but a dramatic increase in H3K36me3. Furthermore, KDM2A recruits the transcription factor E2F1 and its co-factor HCFC1 to the promoters of key genes required for meiosis entry and progression, such as *Stra8*, *Meiosin*, *Spo11*, and *Sycp1*. Collectively, our study unveils an essential role for KDM2A in mediating H3K36me2/3 deposition and controlling the programmed gene expression necessary for the transition from mitosis to meiosis during spermatogenesis.**

**Keywords** KDM2A; Histone Modification; Meiosis; Male Germ Cells; Fertility
**Subject Categories** Cell Cycle; Chromatin, Transcription & Genomics; Development

## Introduction

Meiosis is a specialized process during germ cell development that is accompanied by robust changes in gene expression (Adams and Davies, 2023). At the transition from mitosis to meiosis, retinoic acid (RA) signaling plays a critical role in the remodeling of diploid mitotic spermatogonia into primary spermatocytes by stimulating the expression of STRA8 (stimulated by retinoic acid 8), a pivotal regulator of meiosis (Suzuki et al, 2021). In mice, STRA8 coordinates with its interacting partner MEIOSIN to orchestrate the transition from mitosis to meiosis by initiating meiotic gene activation, which is critical for establishing meiosis-specific chromosomal events (Ishiguro et al, 2020). Recent research has shown that *Znhit1* controls the initiation of meiosis by facilitating the deposition of the histone variant H2A.Z to ensure timely meiotic gene expression (Sun et al, 2022). Moreover, male germ cells also required the incorporation of H3t (a mouse testis-specific histone H3 variant) into the genome during spermatogonial differentiation (Ueda et al, 2017).

Meiosis is initiated during the pre-meiotic S phase, which precedes meiotic prophase I, an extended G2 phase that is essential for numerous meiosis-specific chromosomal events, such as homologous pairing, recombination, synapsis, and chromosome segregation (Shimada and Ishiguro, 2023). Meiotic recombination begins with the formation of hundreds of programmed DNA double-strand breaks (DSBs), catalyzed by SPO11, specifically within 'hotspot' regions (Johnson et al, 2021). Following DSB formation by SPO11 induction, the resected broken DNA ends are then loaded with single-stranded DNA binding proteins (such as DMC1, RPA, and RAD51), which facilitates homology recognition and strand invasion, critical for synapsis initiation through the assembly of the synaptonemal complex (SC), including SYCP1-3, SYCE1-3, and TEX12 (Hinch et al, 2020; Kobayashi et al, 2017; Lan et al, 2020).

Histone methylation is essential for the regulation of chromatin structure and gene expression and is extensively involved in spermatogenesis (McSwiggin and O'Doherty, 2018). In recent years, many histone methyltransferases have been reported to be involved in meiosis (Wang et al, 2017). SUV39H catalyzes the methylation of H3K9 during the transition from spermatogonia to

[1]Institute of Reproductive Health, Tongji Medical College, Huazhong University of Science and Technology, Wuhan 430030, China. [2]Laboratory Animal Center, Huazhong University of Science and Technology, Wuhan 430030, China. [3]College of Animal & Veterinary, Southwest Minzu University, Chengdu 610041, China. [4]Department of Obstetrics and Gynecology, Union Hospital, Tongji Medical College, Huazhong University of Science and Technology, Wuhan 430022, China. [5]Center for Reproduction and Genetics, Department of Obstetrics and Gynecology, The First Affiliated Hospital of USTC, Division of Life Sciences and Medicine, University of Science and Technology of China, Hefei, Anhui 230001, China. [6]Shenzhen Huazhong University of Science and Technology Research Institute, Shenzhen 518057, China. [7]These authors contributed equally: Shenglei Feng, Yiqian Gui, Shi Yin. ✉E-mail: biojxh@ustc.edu.cn; shuiqiaoyuan@hust.edu.cn

spermatocytes, and loss of *Suv39h* in mice causes non-homologous interactions and impaired synapsis, particularly affecting sex chromosomes (Peters et al, 2001). G9A, another mammalian H3K9 methyltransferase, plays a critical role in meiotic homologous recombination, and depletion of G9A in mice disrupts H3K9 methylation and causes failure of spermatogenesis (Takada et al, 2011). In addition, PRDM9 can participate in meiotic recombination initiation in spermatocytes by facilitating the methylation of H3K4, H3K9, and H3K36 (Grey et al, 2018).

Notably, almost all prominently methylated lysine residues on histones have a corresponding demethylase enzyme, and these lysine demethylases (KDMs) are responsible for the reversible removal of histone methylation marks from lysine residues (Dimitrova et al, 2015). Accumulating evidence highlights the role of histone demethylases in male germ cell development and spermatogenesis. For instance, KDM2B, a demethylase that catalyzes the removal of methyl groups from H3K4 and H3K36, plays a crucial role in regulating the proliferation of spermatogonia and ensuring the long-term sustainability of spermatogenesis in mice (Ozawa et al, 2016). KDM3A acts as a demethylase targeting H3K9 and plays a key role in spermiogenesis by controlling the expression of genes such as *Tnp1* and *Prm1* (Okada et al, 2007). KDM3B contributes to the activation of certain genes crucial for spermatogonial stem cell maintenance by demethylating H3K9 during the transition from prospermatogonia to spermatogonia (Kuroki et al, 2020). Although some histone demethylases have been identified as important factors at different stages of spermatogenesis, whether and how they are required for meiosis, in particular meiotic initiation, remains unknown.

In this study, we identify KDM2A, an H3K36-specific lysine demethylase enzyme (Liu et al, 2021), as essential for meiotic initiation and progression. Germ cell-specific ablation of *Kdm2a* causes complete sterility, with spermatogenesis ultimately arrested at the zygotene stage of meiosis. RNA-seq analysis reveals that KDM2A can repress the expression of genes essential for spermatogonial development and facilitate the expression of genes required for meiosis. Strikingly, CUT&RUN-seq analyses reveal a specific role for KDM2A in the removal of H3K36me3 from the target genomic region in male germ cells, suggesting that KDM2A may act as a lysine demethylase towards H3K36me3 in germ cells, distinct from its traditional functions as an H3K36me2 demethylase in somatic cells (Liu et al, 2021). Furthermore, we found that KDM2A can recruit the transcriptional factor E2F1 and its co-factor HCFC1 to the promoters of key genes (e.g., *Stra8, Meiosin, Spo11*, and *Sycp1*) that are required for meiosis and control their transcriptional expression. Our findings shed light on the crucial role of the demethylase KDM2A for meiotic progress and provide new insights into the molecular mechanism orchestrating the programmed gene expression during the mitosis-to-meiosis transition.

## Results

### Spatiotemporal expression of KDM2A and H3K36me1/2/3 distribution during spermatogenesis

To understand the function of KDM2A in male germ cell development, we first determined its expression pattern during spermatogenesis. The results showed that both the mRNA and protein expression levels of

*Kdm2a* were highly expressed in the testes (Fig. EV1A,B). Interestingly, the abundance of mRNA and protein of *Kdm2a* in mouse testes increased sharply at postnatal day 14 (P14) and then gradually decreased (Fig. EV1C,D). We then reanalyzed the published single-cell RNA sequencing data from adult and P7 testes (Hermann et al, 2018; Wang et al, 2019) to further investigate the expression pattern of *Kdm2a* during spermatogenesis. The results showed that *Kdm2a* was expressed at high levels in both spermatogonia and spermatocytes, but was almost absent in spermatids and somatic cells (Fig. 1A). In the juvenile (P7) mice, *Kdm2a* was predominantly expressed in Sertoli cells and spermatogonia (especially in earlier spermatogonia) (Fig. 1B,C). Interestingly, further immunofluorescence (IF) assays revealed that KDM2A was highly expressed in spermatogonia, spermatocytes, and Sertoli cells but not in round and elongating spermatids (Fig. 1D). Subsequently, we performed whole-mount staining of seminiferous tubules from adult testes with antibodies against PLZF (also called ZBTB16, a marker for undifferentiated spermatogonia) and c-KIT (a marker for differentiated spermatogonia), and found that KDM2A was clearly expressed in all types of undifferentiated spermatogonia including As (single), A$_{pr}$ (paired), and A$_{al}$ (aligned), and differentiated spermatogonia (Fig. EV1E,F). To characterize the expression profile of KDM2A in the first wave of spermatogenesis, we checked the co-localization of KDM2A with PLZF, SALL4 (another marker of undifferentiated spermatogonia) or c-KIT in P10 mouse testes. Similarly, KDM2A was also highly expressed in various types of juvenile spermatogonia (Figs. 1E,F and EV1G). We then examined the detailed localization of KDM2A during meiosis through chromosome spread stained with antibodies against SYCP3 (synaptonemal complex protein 3). We found that KDM2A expression was high at the leptotene and zygotene stages, but decreased from the pachytene stage onwards (Fig. 1G). These results suggest that KDM2A may play a role during the developmental window from mitotic spermatogonia to meiotic spermatocytes.

Since KDM2A is considered the main histone H3K36 demethylase in mammals (Liu et al, 2021), we next characterized the distribution of H3K36me1, H3K36me2, and H3K36me3 in male germ cells. Interestingly, these three histone makers exhibit distinct expression patterns in male germ cells (Fig. EV1H–L). Especially, H3K36me1 and H3K36me3 displayed a weak signal in juvenile (P14) and adult spermatogonia (Fig. EV1H,J,L) but clearly located in the nucleus of spermatocytes (Fig. EV1I); conversely, H3K36me2 was highly expressed in both P14 and adult spermatogonia (Fig. EV1H,K) but poorly distributed in P14 spermatocytes (Fig. EV1I). To further determine the exact expression of H3K36me1/2/3 at which stage of the spermatocyte, we performed a chromosome spreading assay on different stages of spermatocytes. The results showed that H3K36me1 and H3K36me2 were highly expressed in leptotene and zygotene spermatocytes, but showed low expression in pachytene and diplotene spermatocytes (Fig. EV1I), which is consistent with expression pattern of KDM2A in spermatocytes. In contrast, the expression level of H3K36me3 was significantly higher in pachytene and diplotene spermatocytes compared to the early meiotic stages (Fig. EV1I).

### KDM2A is required for spermatogenesis and male fertility

To determine the physiological functions of KDM2A in male germ cell development and spermatogenesis, we generated germline conditional *Kdm2a* knockout mice by using the *Stra8-GFPCre*

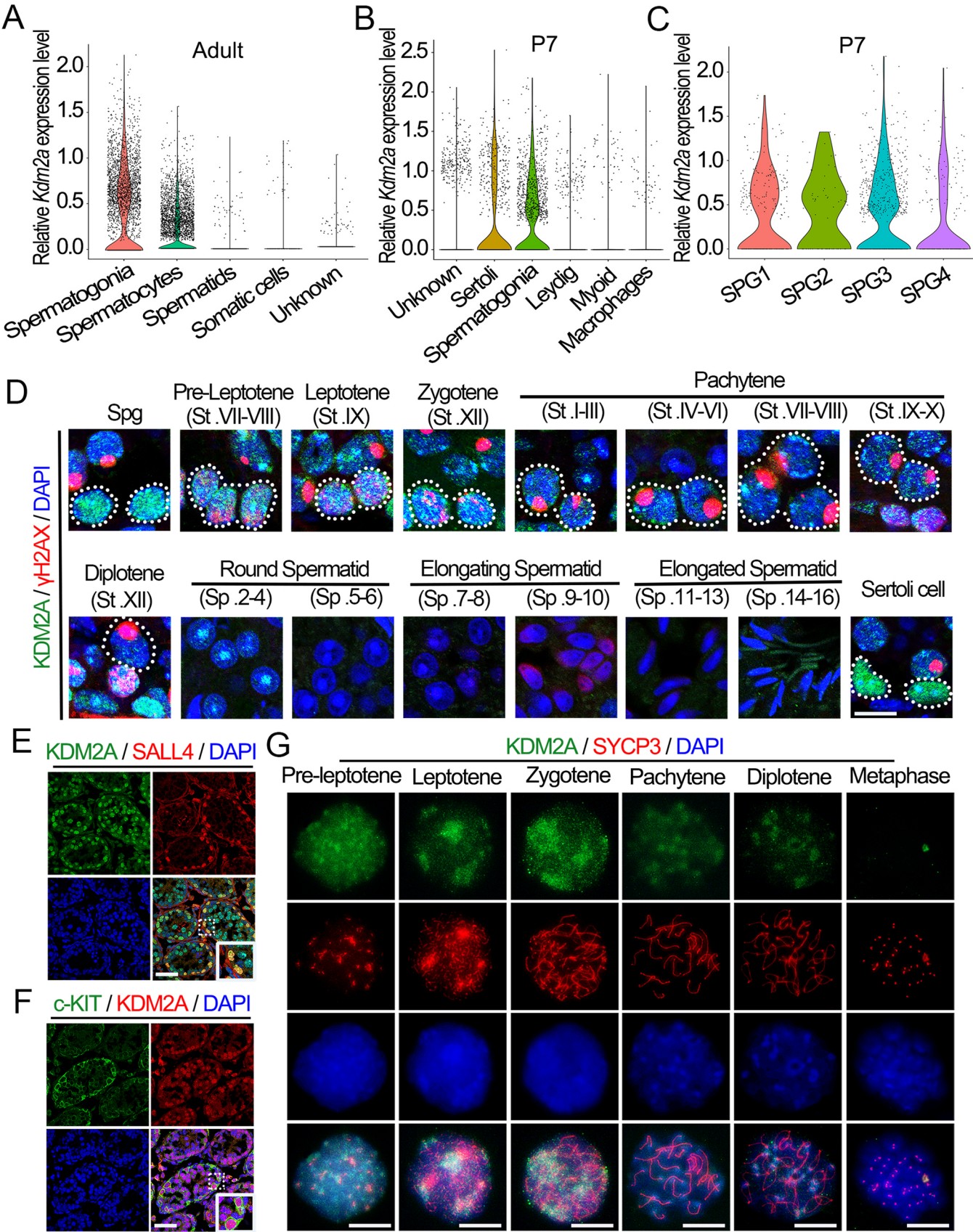

◄  **Figure 1.  KDM2A displays a dynamic expression pattern during spermatogenesis.**

(A–C) Scatter plots showing the expression of *Kdm2a* in different types of testicular cell populations based on previously reported single-cell RNA-seq data in adult (A) and P7 mouse testes (B, C). SPG1, 2, 3, 4 correspond to SSCs (spermatogenic stem cells), undifferentiated spermatogonia, early differentiating spermatogonia and late differentiating spermatogonia, respectively. (D) Anti-KDM2A and anti-γH2AX antibodies were used for double immunostaining of wild-type (WT) testicular cells from adult testicular cryosections. Spotted lines show the indicated cell type. Nuclei were stained with DAPI. Scale bars = 10 µm. (E, F) Co-immunofluorescence staining of SALL4 (E) and c-KIT (F) with KDM2A in WT testis sections from P10 mice are shown. Nuclei were stained with DAPI. Scale bars = 50 µm. (G) Double immunostainings with KDM2A and SYCP3 on surface-spread spermatocytes from WT P21 mice are shown. Nuclei were stained with DAPI. Scale bars = 5 µm. Biologically independent mice (*n* = 3) were examined in three separate experiments (D–G). Source data are available online for this figure.

knock-in mice to delete exon 6 of the *Kdm2a* gene in germ cells (*Kdm2a*$^{flox/Del}$; *Stra8-GFPCre*, hereafter called *Kdm2a* cKO) (Fig. EV2A). *Stra8*-GFPCre induces recombination from type A1 spermatogonia as early as P3 in males (Lin et al, 2017; Sadate-Ngatchou et al, 2008). Western blot (WB) and IF analysis showed a significant decrease in KDM2A protein levels and a complete absence of KDM2A in germ cells of adult (P60) and juvenile (P5 and P8) *Kdm2a* cKO mouse testes compared to controls (*Kdm2a*$^{flox/flox}$ or *Kdm2a*$^{flox/+}$, hereafter called control) (Figs. 2A,B and EV2B), indicating the successful generation of a germline-specific *Kdm2a* knockout mouse model. Although *Kdm2a* cKO males maintained normal body weight and were viable, they were completely infertile in fertility tests and their testes were significantly smaller than those of littermate controls (Fig. 2C,D). Consistent with the reduction in testis size, histological analysis of *Kdm2a* cKO males revealed impaired spermatogenesis as evidenced by the absence of post-meiotic spermatids and mature spermatozoa in the seminiferous tubules and cauda of the epididymis (Fig. 2E). Interestingly, the testis/body weight ratio of *Kdm2a* cKO males decreased significantly from P14 onwards (Fig. 2F), suggesting that the primary defect had already occurred during meiotic prophase and even early. Further IF analysis of c-KIT$^+$ cells (differentiating spermatogonia) in control and *Kdm2a* cKO testes at P10 revealed a comparable number of c-KIT$^+$ cells between control and *Kdm2a* cKO testes (Fig. 2G,H), suggesting that KDM2A depletion does not affect spermatogonial differentiation in the first wave of spermatogenesis. We then co-stained c-KIT with STRA8 in P10 testes and found a significant decrease in the number of spermatocytes (STRA8$^+$c-KIT$^-$ cells represent preleptotene spermatocytes) in *Kdm2a* cKO testes compared to controls, but no difference in the number of early (STRA8$^+$c-KIT$^+$ cells represent A1 to A4) and late (STRA8$^-$c-KIT$^+$ cells represent intermediate to B-type) differentiating spermatogonia was observed between control and *Kdm2a* cKO testes (Fig. EV2C–E), demonstrating that KDM2A knockout in germ cells can affect meiotic entry but not spermatogonial differentiation during the first round of spermatogenesis. In addition, we performed IF assays by co-staining PLZF with STRA8 in P10 and P14 testes to dissect spermatogonia fate and meiotic entry in *Kdm2a* cKO mice. The results showed that although PLZF$^+$STRA8$^+$ spermatogonia appeared normal in *Kdm2a* cKO testes, the number of PLZF$^-$STRA8$^+$ preleptotene spermatocytes in the seminiferous tubules was significantly reduced compared to littermate controls (Fig. 2I–L), confirming that spermatogonial differentiation was not affected in the first wave of spermatogenesis and meiotic entry was impaired in *Kdm2a* cKO males. Interestingly, although the number of PLZF$^+$ undifferentiated spermatogonia was not affected in adult *Kdm2a* cKO testes (Fig. EV2F,G), the number of c-KIT$^+$ spermatogonia was reduced in adult *Kdm2a*

cKO testes compared to controls (Fig. EV2H,I), suggesting that KDM2A deletion may affect the differentiation of adult spermatogonia. To further verify this, we generated *Kdm2a*$^{flox/flox}$; *Ddx4*-Cre$^{ERT2}$ male mice for tamoxifen-induced KDM2A deletion in germ cells (referred to as iKO) to investigate whether the spermatogonial differentiation process is affected by *Kdm2a* ablation in adulthood (Fig. EV3A). Ablation of KDM2A in the c-KIT$^+$ spermatogonia of iKO mice was confirmed by IF staining after tamoxifen treatment (day 35) in 8-week-old adult mice (Fig. EV3B). Notably, the iKO mice exhibited a decrease in the number of c-KIT$^+$ spermatogonia and a comparable number of PLZF$^+$ spermatogonia, which phenocopied the adult *Kdm2a* cKO mice (Fig. EV3D–I), further demonstrating that KDM2A is essential for spermatogonial differentiation during steady-state adult spermatogenesis. Together, these data indicate that KDM2A is required for male germ cell development and male fertility.

## KDM2A is crucial for meiosis initiation and progression

To further investigate the defective meiosis in *Kdm2a* cKO males, we performed immunostaining for SYCP3, a synaptonemal complex (SC) protein and γH2AX, a marker for DNA double-strand break (DSB) formation. Compared with controls, the number of tubules that have PLZF$^-$/STRA8$^+$/rH2AX$^+$ and PLZF$^-$/STRA8$^+$/SYCP3$^+$ pre-leptotene spermatocytes were significantly decreased in *Kdm2a* cKO mice, demonstrating KDM2A was required for meiotic initiation (Fig. 3A–D). In addition, the number of SYCP3$^+$/γH2AX$^+$ spermatocytes in P14 *Kdm2a* cKO mice was significantly reduced and even absent in some seminiferous tubules (Fig. EV4A–C). To evaluate the abnormality of *Kdm2a* cKO spermatocytes during meiosis, we examined the meiotic prophase I by staining SYCP3 (SC lateral elements) and γH2AX in chromosome spread nuclei of spermatocytes. In the most advanced *Kdm2a* cKO spermatocytes, the chromosome axial elements were well formed but synapsis was abnormal, and these cells were referred to as zygotene-like spermatocytes (Figs. 3E,F and EV4D). Similarly, while SYCP1 (a marker of SC central filaments) marked the synapsed regions in control spermatocytes, the zygotene-like spermatocytes in *Kdm2a* cKO either lacked SYCP1 signal or exhibited abnormal accumulation in sister chromatids (Fig. 3G), showing that KDM2A was required for chromosomal synapsis. Interestingly, we noticed that γH2AX signals in *Kdm2a* cKO leptotene and zygotene-like spermatocytes were weaker than that in control (Fig. 3F), suggesting that DSB formation was partially compromised in *Kdm2a* cKO mice. To further evaluate the process of meiotic recombination, we examined two DNA recombinase markers, RAD51 and DMC1, which could drive strand invasion into homologous DNA duplex during meiotic

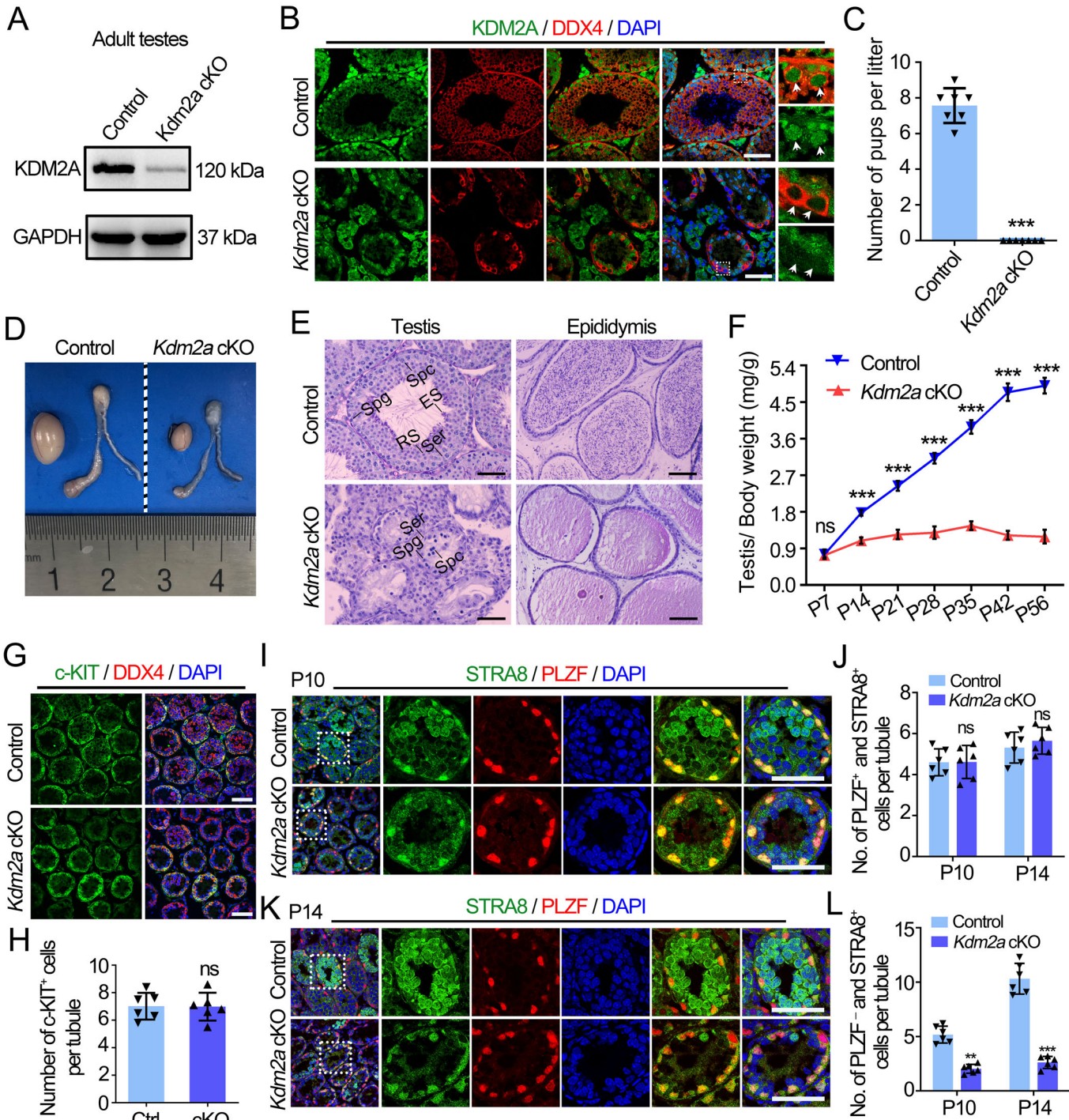

prophase I. The number of RAD51 foci and DMC1 foci were dramatically decreased in *Kdm2a* cKO leptotene and zygotene-like spermatocytes, suggesting a defect in early meiotic recombination in *Kdm2a* cKO spermatocytes (Figs. 3H,I and EV4E,F). In addition, the TUNEL assay revealed a higher proportion of apoptotic cells from the early P10 stage in *Kdm2a* cKO testes than in controls (Fig. EV4G,H). Taken together, these results indicate that KDM2A plays a critical role in meiotic initiation and progression during the first wave of spermatogenesis.

## Loss of *Kdm2a* causes transcriptional dysregulation in male germ cells

To clarify the potential regulatory mechanism of KDM2A in male meiosis, we performed RNA-seq analysis using purified c-KIT⁺ spermatogonia from control and *Kdm2a* cKO mice at P10 (when the initial wave starts at the meiotic entry) (Fig. 4A). IF analysis confirmed that the purity of c-KIT⁺ cells was above 90% in both control and *Kdm2a* cKO groups (Fig. EV5A), and no significant

**Figure 2. Conditional knockout of *Kdm2a* in germ cells causes male sterility.**

(A) Western Blotting (WB) analyses of KDM2A protein levels in control and *Kdm2a* cKO adult testes using GAPDH as a loading control. (B) Immunofluorescence (IF) staining of KDM2A in control and *Kdm2a* cKO testes of adult mice. DDX4 was co-stained to indicate the location of the germ cell. Enlarged images are shown in right panels. Arrowheads indicate KDM2A signals in the DDX4 positive cells. The DNA was stained with DAPI. Scale bars = 50 μm. (C) Histograms show the average number of pups per litter when control and *Kdm2a* cKO adult male mice were mated with WT female mice. Data are presented as mean ± SD, *n* = 7. *P* value was calculated using a two-tailed Student's *t*-test. ***$P < 0.0001$. (D) Gross morphology of testes and epididymides from adult control and *Kdm2a* cKO mice. (E) Periodic acid-Schiff (PAS) staining of testes and epididymides from control and *Kdm2a* cKO adult mice. Spg: spermatogonia, Ser: Sertoli cells, Spc: spermatocytes, RS: round spermatids, ES: elongated spermatids. Scale bars = 50 μm. (F) Testis growth curves of control and *Kdm2a* cKO mice from postnatal day 7 (P7) to P56. Data are presented as mean ± SD, *n* = 3 (three biological replicates). *P* value was calculated using a two-tailed Student's *t*-test. ns, not significant. ***$P < 0.0001$. (G) Co-immunofluorescent staining of c-KIT (green) with DDX4 (red) on testis sections from control and *Kdm2a* cKO mice at P10. Scale bars = 50 μm. (H) Quantification of c-KIT$^+$ cells per tubule for (G). Data are presented as mean ± SD, *n* = 6 biological replicates. *P* value was calculated using a two-tailed Student's *t*-test. ns, not significant. (I) Co-immunofluorescent staining of STRA8 (green) with PLZF (red) on testis sections from control and *Kdm2a* cKO mice at P10. Scale bars = 50 μm. (J) Quantification of PLZF$^+$ and STRA8$^+$ cells per tubule for (I). Data are presented as mean ± SD, *n* = 6 biological replicates. *P* value was calculated using a two-tailed Student's *t*-test. ns, not significant. (K) Co-immunofluorescent staining of STRA8 (green) with PLZF (red) on testis sections from control and *Kdm2a* cKO mice at P14. Scale bars = 50 μm. (L) Quantification of PLZF$^-$ and STRA8$^+$ cells per tubule for (K). Data are presented as mean ± SD, *n* = 6 biological replicates. *P* value was calculated using a two-tailed Student's *t*-test. **$P = 0.0045$, ***$P = 0.0008$. All images are representative of *n* = 3 mice per genotype. Source data are available online for this figure.

difference was observed in the number of STRA8$^+$c-KIT$^+$ (A1 to A4) and STRA8$^-$ c-KIT$^+$ (intermediate to B type) cells between control and *Kdm2a* cKO mice (Fig. EV5B,C), suggesting that the c-KIT$^+$ enriched cells derived from control and *Kdm2a* cKO P10 testes are comparable and suitable for subsequent transcriptome analyses. A total of 3856 differentially expressed genes (DEGs), including 1934 upregulated and 1922 downregulated genes, were identified from the RNA-seq data in *Kdm2a* cKO mice (Fig. 4B; Dataset EV1). Interestingly, *Kdm2a* did not appear in the DEGs and showed unchanged mRNA levels in the *Kdm2a* cKO cells by qPCR assay (Fig. EV5D); however, it showed exon-6 skipping based on the RNA-seq data viewed in the Integrative Genomic Viewer browser (Fig. EV5E). By RT-PCR analysis, we found that the *Kdm2a* transcript with the exon-6 deletion was still present in *Kdm2a* cKO germ cells (Fig. EV5F). These data suggest that the Cre-mediated deletion of exon-6 of *Kdm2a* did not lead to degradation of this truncated transcript, but likely resulted in the production of non-functional proteins that were subsequently degraded. By joint analysis of the DEGs with previously published scRNA-seq data from spermatogenic cells (Hermann et al, 2018), we found that the upregulated genes in *Kdm2a* cKO germ cells were overall expressed in early pseudotime (Fig. 4C). In contrast, the downregulated genes in *Kdm2a* cKO germ cells were found around the mid to later stages of pseudotime (Fig. 4D). This conclusion was further supported by a reanalysis of the data from previous studies on transcriptomes during spermatogenesis (da Cruz et al, 2016; Wang et al, 2019). Specifically, the upregulated genes in *Kdm2a* cKO germ cells were highly expressed in early spermatogonia and the expression declined from late spermatogonia and prophase spermatocytes (Figs. 4E and EV5G), whereas the downregulated genes in *Kdm2a* cKO germ cells were overall less expressed in spermatogonia and highly expressed in prophase spermatocytes (Fig. 4F). In addition, the expression levels of downregulated genes in control c-KIT$^+$ cells were significantly higher than those of upregulated genes (Fig. EV5H), and the fold changes of downregulated genes were much greater than those of upregulated genes (Fig. EV5I), suggesting that KDM2A tends to regulate these downregulated genes. Gene Ontology (GO) analysis revealed that the upregulated genes in the *Kdm2a* mutant were involved in 'negative regulation of cell differentiation', 'cell fate commitment', and 'developmental biology' (Fig. 4G), while the downregulated genes participated in many aspects of meiosis, such as 'homologous

recombination', 'synaptonemal complex assembly', 'double-strand break repair' and 'meiotic telomere clustering' (Fig. 4H). Interestingly, many of the upregulated genes were involved in SSCs (spermatogonial stem cells)) maintenance and differentiation, whereas some of the downregulated genes were associated with meiotic initiation (Fig. 4I). In addition, 10 upregulated genes related to spermatogonial development and 10 downregulated genes related to meiotic process were verified in c-KIT$^+$ purified germ cells from P10 testes by RT-qPCR assay (Appendix Fig. S1A,B). Notably, we found a significant change in the protein expression levels of some meiosis-related genes (e.g., *Sycp1*, *Dmc1*, *Stra8*, *Rec8*, and *Syce1*) in *Kdm2a* cKO purified germ cells compared to controls (Appendix Fig. S1C,D). These data suggest that ablation of *Kdm2a* causes both transcriptional and translational dysregulation in male germ cells, particularly for meiosis-related gene expression.

In line with the observed defects in meiotic initiation, gene set enrichment analysis (GSEA) revealed that deletion of *Kdm2a* significantly reduced the enrichment of upregulated genes associated with meiosis. *Kdm2a* deletion led to a remarkably decreased enrichment of upregulated genes in meiosis (Fig. 4J). Notably, we found that the expression of the transcriptional activator *Stra8* and its cofactor *Meiosin*, which facilitates the activation of the meiotic transcriptional program, were both dramatically downregulated in *Kdm2a* cKO germ cells (Fig. 4I; Appendix Fig. S1B). Consistently, GSEA analysis revealed that either *Stra8* or *Meiosin* target genes were downregulated in *Kdm2a* cKO testes (Fig. 4K,L), further suggesting a critical role of KDM2A in orchestrating the meiotic initiation by facilitating the expression of *Stra8* and *Meiosin*. Combining with the published RNA-seq data of *Stra8* (Kojima et al, 2019) and *Meiosin* (Ishiguro et al, 2020), we found that 433 upregulated genes and 479 downregulated genes in *Stra8* mutant testes overlapped with the DEGs of *Kdm2a* cKO mice (Appendix Fig. S1E,F), while the *Meiosin* and *Kdm2a* shared 94 upregulated genes and 250 downregulated genes in their respective mutants (Appendix Fig. S1G,H). Moreover, a high correlation (R = 0.64, *p* < 2.2e−16) of fold-change values of shared DEGs between *Kdm2a* cKO germ cells and *Stra8* KO testes (Appendix Fig. S1I,J) suggested that *Kdm2a* and *Stra8* has a functional association in inhibiting gene expression in addition to enhancing gene expression. However, a relatively low correlation (R = 0.26, *p* < 4e−10) between *Kdm2a* cKO germ cells and *Meiosin* KO testes was observed due to

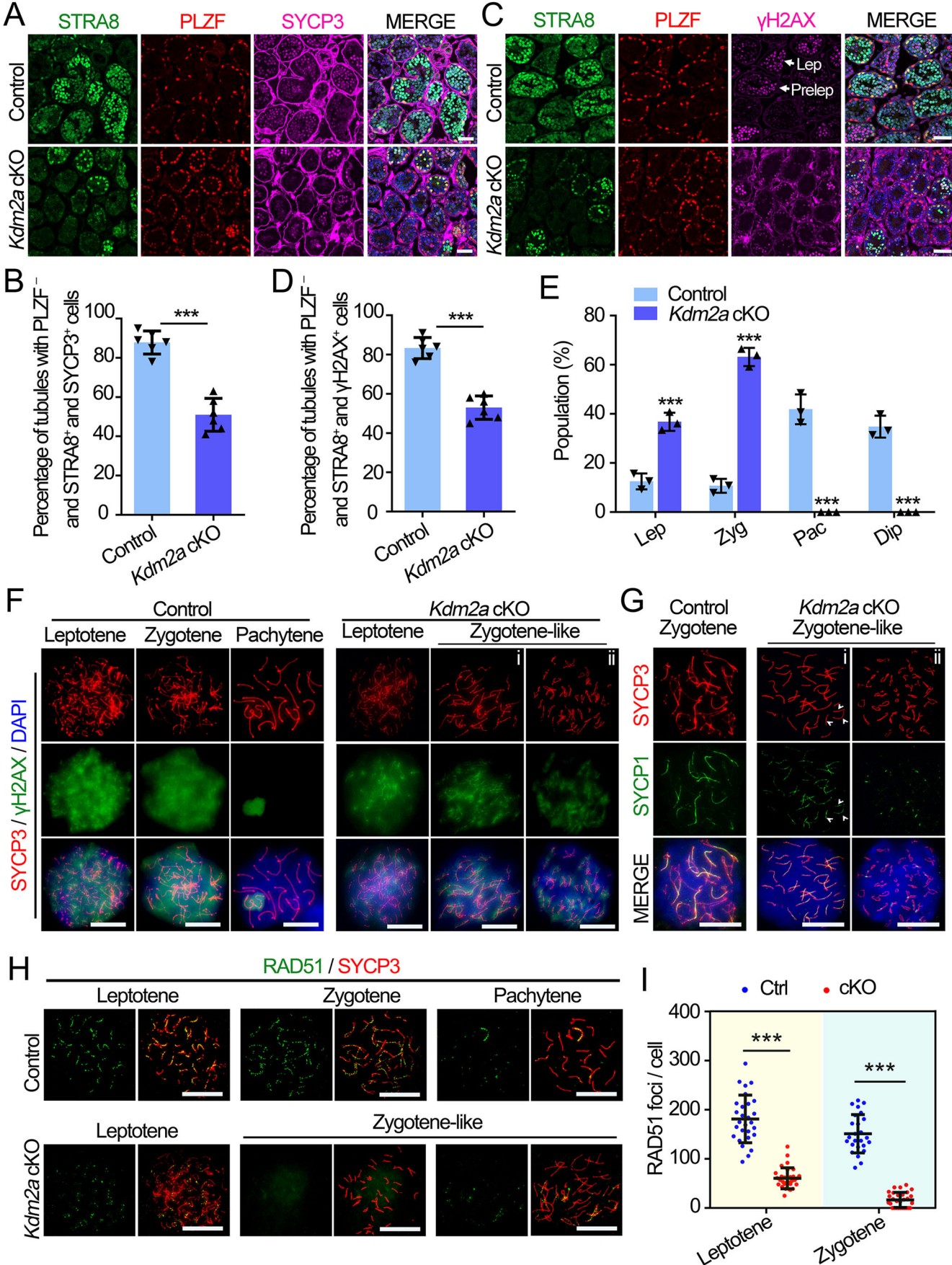

◄ **Figure 3. KDM2A is required for progression of male meiosis.**

(A) Co-immunostaining of STRA8 (green), PLZF (red), and SYCP3 (purple) in testis sections from control and *Kdm2a* cKO mice at P10. The DNA was stained with DAPI. Scale bars = 50 µm. (B) Quantification of the number of seminiferous tubules that have PLZF⁻, STRA8⁺, and SYCP3⁺ cells per total seminiferous tubules for (A). Data are presented as mean ± SD, n = 6 biological replicates. *P* value was calculated using a two-tailed Student's *t*-test. ***P < 0.0001. (C) Immunostaining of STRA8 (green), PLZF (red), and γH2AX (purple) in testis sections from control and *Kdm2a* cKO mice at P10. The DNA was stained with DAPI. Lep, leptotene spermatocytes. Prelep, preleptene spermatocytes. Scale bars = 50 µm. (D) Quantification of the number of seminiferous tubules that have PLZF⁻, STRA8⁺, and γH2AX⁺ cells per total seminiferous tubules for (C). Data are presented as mean ± SD, n = 6 biological replicates. *P* value was calculated using a two-tailed Student's *t*-test. ***P < 0.0001. (E) Percentages of spermatocytes at the leptotene (Lep), zygotene (Zyg), pachytene (Pac), and diplotene (Dip) stages from control and *Kdm2a* cKO mice at P21. Data are presented as mean ± SD. n = 3 biological replicates. *P* value was calculated using a two-tailed Student's *t*-test. left to right: ***P = 0.001, ***P < 0.0001, ***p = 0.0002, ***P = 0.0002. (F) Co-immunofluorescent staining of SYCP3 with γH2AX in spermatocyte chromosome spreads from control and *Kdm2a* cKO mice at P18. Nuclei were stained with DAPI. Scale bars = 5 µm. (G) Chromosome spreads of control zygotene spermatocytes and *Kdm2a* cKO zygotene-like cells at P18 were stained for SYCP3 and SYCP1. Scale bars = 5 µm. Abnormal SYCP1 signals accumulated in unsynapsed sister chromatids in *Kdm2a* cKO mice are indicated by the white arrow. (H) Spermatocyte spreads from control and *Kdm2a* cKO testes at P18 were stained for SYCP3 and RAD51. Scale bars = 5 µm. (I) Scatter plot showing the numbers of RAD51 foci per cell on SYCP3 axes in leptotene and zygotene spermatocytes for (H). Data are presented as mean ± SD. A total of n = 28 control leptotema, n = 30 *Kdm2a* cKO leptotema, n = 25 control zygonema, and n = 29 *Kdm2a* cKO zygonema were counted from three biologically independent mice for each genotype. *P* value was calculated using a two-tailed Student's *t*-test. ***P < 0.0001. Source data are available online for this figure.

their limited common downregulated genes. These bioinformatic data indicated that KDM2A may maintain a close functional relationship with *Stra8* and *Meiosin* in promoting the expression of meiosis-related genes, highlighting a crucial role of KDM2A in controlling meiotic gene expression by upregulating *Stra8* and *Meiosin*.

## KDM2A binds to the genes associated with meiosis

As a histone-modifying enzyme, KDM2A plays a crucial role in the regulation of transcription by modulating the chromatin state. Therefore, we further investigated the KDM2A target sites on the genome by Cleavage Under Targets & Release Using Nuclease followed by sequencing analyses (CUT&RUN-seq). As a result, KDM2A bound to 24,689 sites (14,086 nearest genes), of which 44% and 34.6% resided around the transcriptional start site and gene promoter (TSS promoter) and the intergenic genome regions, respectively (Fig. 5A; Dataset EV2). Combined with our RNA-seq data, most of the DEGs (1251 upregulated and 1107 downregulated) in *Kdm2a* cKO germ cells were found to be bound by KDM2A (Fig. 5B). GO term analysis indicated that these upregulated genes in *Kdm2a* cKO germ cells that were bound by KDM2A mainly participated in 'negative regulation of cell differentiation' and 'cell fate commitment' (Fig. 5C), whereas the downregulated genes in *Kdm2a* cKO germ cells bound by KDM2A were involved in 'homologous chromosome pairing', 'synaptonemal complex assembly', 'meiotic telomere clustering', and 'meiotic DSB formation' (Fig. 5D). Together, these results suggest that KDM2A could bind to a wide range of meiosis-related genes.

## *Kdm2a* ablation leads to abnormal levels of H3K36me2/3 in germ cells

To investigate whether KDM2A functions as a demethylase targeting histone H3k36me2, as expected from previous studies(Liu et al, 2021), we examined the expression of H3K36me1/2/3 in *Kdm2a* cKO testes. The IF result showed that H3K36me1 was not affected by *Kdm2a* ablation (Fig. 5E; Appendix Fig. S2A). Surprisingly, H3K36me2 showed a decreased IF signal in the c-KIT⁺ cells (differentiated spermatogonia) and SYCP3⁺ cells (spermatocytes) of *Kdm2a* cKO testes in comparison to the controls (Fig. 5F; Appendix Fig. S2B), while H3K36me3 displayed

an opposite feature with abnormally enhanced expression in *Kdm2a* cKO testes (Fig. 5G; Appendix Fig. S2C). Specifically, the expression of H3K36me2 showed a decrease in leptotene and zygotene spermatocytes of the *Kdm2a* cKO mice, whereas H3K36me3 had an apparently increased distribution compared to controls (Appendix Fig. S2D,E). Furthermore, the results of the WB assay using purified c-KIT⁺ cells confirmed this phenotype, showing a higher expression level of H3K36me3 and relatively lower expression of H3K36me2 in *Kdm2a* cKO mice compared with that in controls (Fig. 5H). These data exceeded our expectations and suggest that KDM2A likely acts as a demethylase, specifically targeting H3K36me3 but not H3K36me2 in the testes.

To better gain mechanistic insights into the meiosis defects caused by *Kdm2a* deletion, we set out to investigate the genome-wide distribution of H3K36me2/3 by CUT&RUN-seq in the c-KIT⁺ cells from the P10 control and *Kdm2a* cKO mice. Our CUT&RUN-seq analyses identified a total of 24,337 H3K36me2 peaks and 22,319 H3K36me3 peaks, which cover 11,712 and 10,937 genes in c-KIT⁺ germ cells, respectively (Fig. 5I; Dataset EV3). However, in the *Kdm2a* cKO germ cells, the number of peaks of H3K36me2 decreased to 8510, while the H3K36me3 peaks increased to 65535 (Fig. 5I; Dataset EV3). Consistent with our hypothesis of KDM2A likely being the H3K36me3 methyltransferase during spermatogenesis, we found that the *Kdm2a* cKO c-KIT⁺ germ cells showed a reduction of H3K36me2 level but a drastic increase of H3K36me3 level (Fig. 5J,K).

To determine whether KDM2A controls the H3K36me2/3 deposition on its target genes, we analyzed the change of H3K36me2/3 level on KDM2A bound genes. Interestingly, we found most of the genes with increased H3K36me3 signal (6366/8756) and with decreased H3K36me2 signal (6371/7786) were both overlapped with the KDM2A target genes (Fig. 5L,M), although these two group genes were not well overlapped (Appendix Fig. S2F). Because H3K36me3 is commonly known to play a role in transcriptional activation (Sun et al, 2020), but in some cases, it has an inhibitory effect on transcription (Ballare et al, 2012). We then compared these genes with the DEGs and found that 633 out of 1251 genes that were upregulated in *Kdm2a* cKO germ cells and bound by KDM2A overlapped with those genes with increased H3K36me3 occupancy in the *Kdm2a* cKO cells (Fig. 5N). Furthermore, 516 out of 1107 genes that were downregulated in *Kdm2a* cKO germ cells and bound by KDM2A also showed

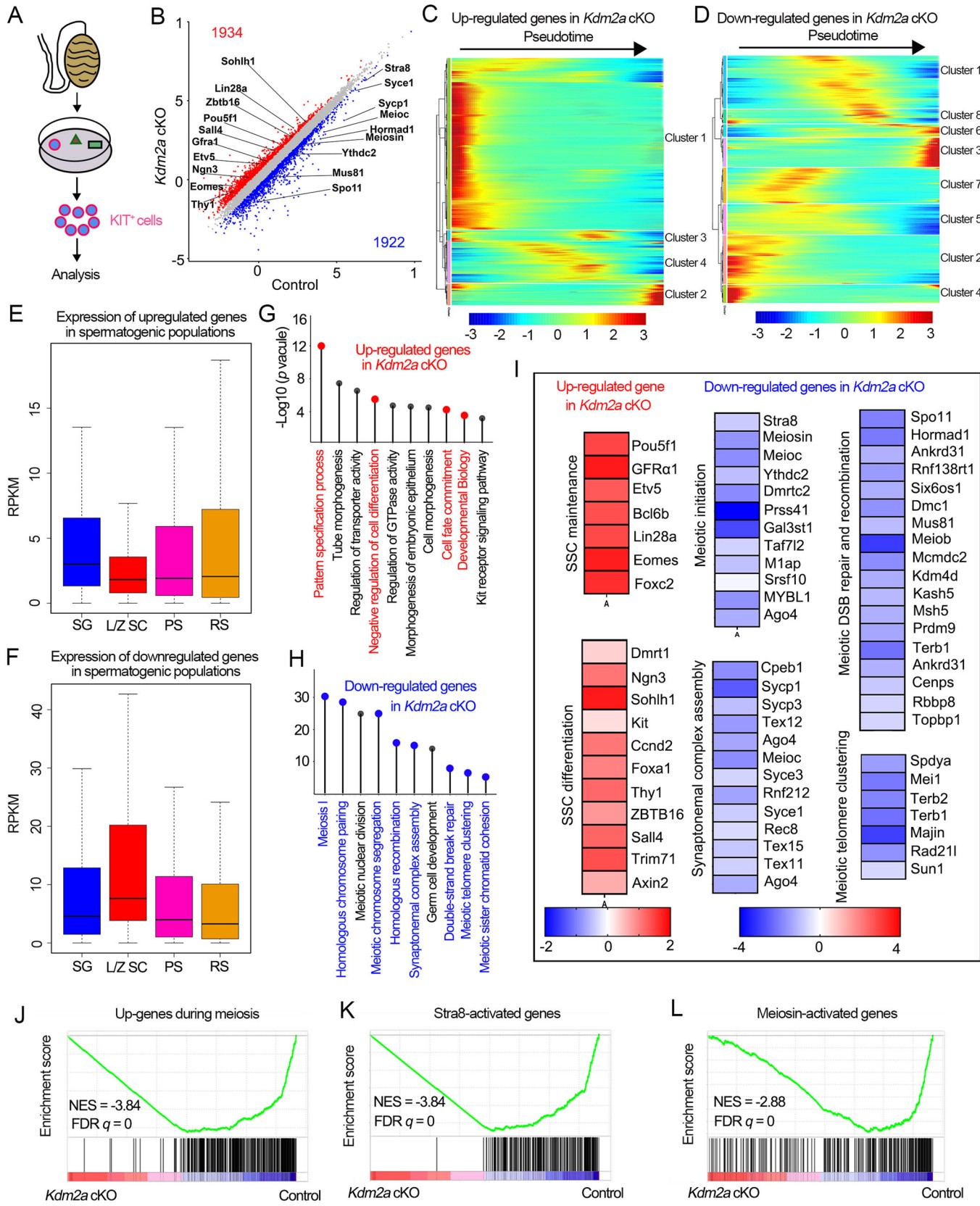

**Figure 4. Transcriptome analysis of *Kdm2a* cKO mice.**

(A) Schematic of the isolation of c-KIT⁺ differentiating spermatogonia cells from P10 control and *Kdm2a* cKO testes. (B) Scatter plot of the transcriptome of c-KIT⁺ cells in control versus *Kdm2a* cKO mice is shown. $n = 3$ (three biological replicates). The numbers of differentially expressed genes (DEGs) are shown ($P < 0.05$, fold change >1.5). Significantly upregulated and downregulated genes in the *Kdm2a* cKO cells are labeled with red and blue color, respectively. DESeq2 Wald test $P$ value was calculated. (C, D) Heatmaps showing the hierarchical relationship among the clusters of DEGs across pseudotime of spermatogenesis. Expressions of the upregulated genes (C) and the downregulated genes (D) in c-KIT⁺ cells of *Kdm2a* cKO mice were assessed by reanalyzing scRNA-seq data of spermatogenic cells. Pseudotime (left to right) corresponds to the developmental trajectory of spermatogenesis (undifferentiated spermatogonia to round spermatids). (E, F) Expression levels (RPKM) of the 1934 upregulated genes (E) and the 1922 downregulated genes (F) in c-KIT⁺ cells of *Kdm2a* cKO mice are shown by box-whisker plot (whiskers indicate min and max. Bounds of box indicate 25th and 75th percentiles quantile with median). The upregulated and downregulated genes in *Kdm2a* cKO were reanalyzed with the previously published data of stage-specific bulk RNA-seq ($n = 3$ biologically independent samples). SG spermatogonia, L/Z SC leptotene/zygotene spermatocytes, PS pachytene spermatocytes, RS round spermatid. (G, H) Gene ontology (GO) analyses of the upregulated genes (G) and downregulated genes (H) in c-KIT⁺ cells of *Kdm2a* cKO mice. Colored terms represent key biological processes related to the development of spermatogenic cells. $P$-values were calculated by using gene ontology functions of HOMER software. (I) Heatmap showing up- and down-regulated genes in *Kdm2a* cKO c-KIT⁺ cells identified by GO analysis. (J–L) GSEA of RNA-seq data for the control and *Kdm2a* cKO c-KIT⁺ cells. Selected gene sets encoded products related to upregulated genes during meiosis (J), *Stra8*-activated genes (K), or *Meiosin*-activated genes (L). $n = 3$ biologically independent samples for each group. NES normalized enriched score. Permutation test on enrichment score was performed following Subramanian algorithm. Nominal $P$-value $= 0$.

increased H3K36me3 occupancy in *Kdm2a* cKO germ cells compared to controls (Fig. 5O). Interestingly, GO analyses revealed that these KDM2A-bound upregulated genes with increased H3K36me3 signal in *Kdm2a* cKO mice were mainly involved in 'negative regulation of cell differentiation' and 'pattern specification process', while the KDM2A-bound downregulated genes with increased H3K36me3 signal in *Kdm2a* cKO mice were closely related with meiotic process such as 'DNA recombination', 'synaptonemal complex assembly' and 'meiotic telomere clustering' (Appendix Fig. S2G,H). Of note, some key genes essential for spermatogonial development (such as *Sall4*, *Lin28a*, *Eomes*, and *Zbtb16*) and meiosis (such as *Stra8*, *Sycp2*, *Hormad1*, and *Syce3*) all have enhanced H3K36me3 signal, especially for their promoter regions (Appendix Fig. S2I,J). These results indicated that the dysregulation of a number of genes caused by KDM2A ablation may be associated with abnormal H3K36me3 deposition in the *Kdm2a* cKO c-KIT⁺ germ cells.

## KDM2A interacts with HCFC1 and E2F1 to regulate target gene expression

To further unravel the molecular function of KDM2A in male germ cell development, we conducted immunoprecipitation (IP) followed by mass spectrometry (MS) using the KDM2A antibody in purified c-KIT⁺ germ cells and detected a total of 118 candidate proteins that were specifically pulled down by the KDM2A (Fig. 6A; Appendix Fig. S3A; Dataset EV4). GO analyses revealed that these proteins were mainly involved in DNA repair and chromatin organization as well as RNA metabolism (such as mRNA splicing and RNA localization) (Fig. 6B). Among the KDM2A-interacting proteins associated with chromatin organization, the key transcription factor E2F1 and its cofactor HCFC1 attracted our attention due to their similar role to KDM2A in cell cycle regulation (Kawakami et al, 2015; Parker et al, 2014), and E2F1 has been reported to play a critical role in spermatogenesis (Jorgez et al, 2021; Rotgers et al, 2015). Through co-immunoprecipitation (Co-IP) assays, we further confirmed that HCFC1 and E2F1 interacted with KDM2A in the testes and this interaction did not depend on the existence of DNA (Fig. 6C). In addition, the KDM2A ablation did not affect the expression of its HCFC1 and E2F1 (Appendix Fig. S3B).

To define whether HCFC1 and E2F1 share similar chromatin binding profiles with KDM2A, we performed CUT&RUN-seq using antibodies against HCFC1 and E2F1, respectively, in purified c-Kit cells. As shown in Fig. 6D,E, both HCFC1 and E2F1 exhibited strong enrichment signals near the TSS region, suggesting a potential role of HCFC1 and E2F1 in transcriptional regulation. Interestingly, most of the KDM2A binding peaks (9520/10,615) accumulated in TSS regions well overlapped with that of HCFC1 (Fig. 6F; Dataset EV5). Similarly, most of the E2F1 binding peaks (8759/10,121) near TSS regions also overlapped well with that of its cofactor HCFC1 (Fig. 6G; Dataset EV5). Furthermore, HCFC1 and E2F1 shared a large number of chromatin binding peaks with KDM2A (Appendix Fig. S3C); thus, it was reasonable that these three proteins shared common binding motifs (Appendix Fig. S3D). Surprisingly, the enrichment signals of both HCFC1 and E2F1 near the TSS regions were abrogated when *Kdm2a* was deletion (Fig. 6D–I), suggesting that KDM2A plays a crucial role in the recruitment of HCFC1 and E2F1 on the target chromatin. As a consequence, when *Kdm2a* was deleted, most of the TSS region with KDM2A binding peaks (8839/10615) showed a sharp decrease in the HCFC1 binding signal (Fig. 6J), and most of the TSS regions with decreased E2F1 occupancy level (4876/5325) were almost accompanied by decreased HCFC1 enrichment signal (Fig. 6K). Combing our RNA-seq, we found that a total of 756 genes upregulated in *Kdm2a* cKO germ cells showed decreased occupancy of both E2F1 and HCFC1 (Fig. 6L), and these genes mainly participated in 'pattern specification process' and 'negative regulation of cell differentiation' (Appendix Fig. S3E). Additionally, a total of 532 downregulated genes have decreased E2F1 and HCFC1 enrichment signal (Fig. 6M), and these genes were predominantly involved in different meiosis events such as 'synaptonemal complex assembly' and 'meiotic recombination' (Appendix Fig. S3F). Notably, the aforementioned genes essential for spermatogonial development (such as *Sall4*, *Lin28a*, *Eomes*, and *Zbtb16*) and some key genes involved in meiosis (such as *Stra8*, *Meiosin*, *Spo11*, and *Sycp1*) showed decreased E2F1 and HCFC1 accumulation signal on their promoter regions (Fig. 6N; Appendix Fig. S3G). These results suggest that abnormal deposition of HCFC1 and E2F1 on TSS regions induced by KDM2A ablation may contribute to the dysregulation of a number of key genes in the *Kdm2a* cKO germ cells.

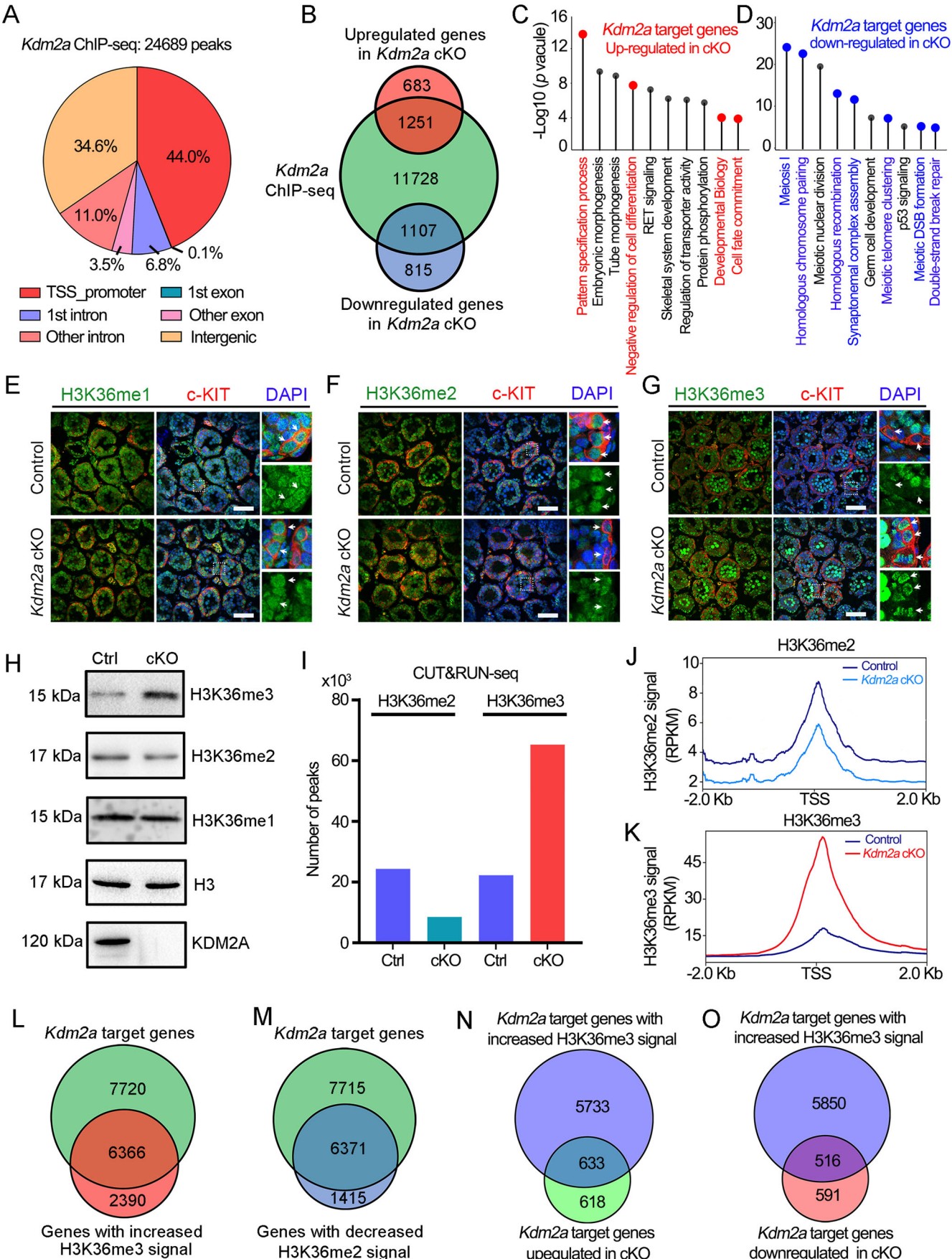

Figure 5.   Ablation of KDM2A in germ cells results in the dysregulation of H3K36me2/3.

(A) Distribution of *Kdm2a* binding sites from *Kdm2a* CUT&RUN-sequencing (CUT&RUN-seq) in c-KIT-positive cells from P10 WT testes. (B) Venn diagram representing the overlap of *Kdm2a* targets genes (CUT&RUN-seq data) and differentially expressed genes (RNA-seq data). (C) GO analyses of the top 10 most enriched biological processes based on their *P*-values combine upregulated genes (our RNA-seq data) with *Kdm2a* target genes (CUT&RUN-seq). *P*-values were calculated by using gene ontology functions of HOMER software. (D) GO analyses of the top 10 most enriched biological processes based on their *P*-values combine downregulated genes (our RNA-seq data) with *Kdm2a* target genes (CUT&RUN-seq). *P*-values were calculated by using gene ontology functions of HOMER software. (E–G) Co-immunofluorescent staining of H3K36me1 (E), H3K36me2 (F), and H3K36me3 (G) with c-KIT in testis sections from control and *Kdm2a* cKO mice at P10 are shown, respectively. The DNA was stained with DAPI. Enlarged images are shown on the right panel. Arrows indicate c-KIT$^+$ spermatogonia. Scale bars = 50 μm. (H) WB analyses the expression of H3K36me1, H3K36me2, and H3K36me3 in c-KIT$^+$ cells isolated from control and *Kdm2a* cKO mice at P10. H3 was served as loading control. (I) Barplot of the peaks distribution of H3K36me2 and H3K36me3 CUT&RUN-seq c-KIT$^+$ cells isolated from control and *Kdm2a* cKO mice at P10. (J, K) Average H3K36me2 (J) or H3K36me3 (K) CUT&RUN signals of control and *Kdm2a* cKO c-KIT-positive cells within −2 kb/+2 kb of TSS. (L, M) Venn diagram depicting the overlap of *Kdm2a* bound genes in WT c-KIT-positive cells and genes with increased H3K36me3 signal (L) or genes with decreased H3K36me2 signal (M) in *Kdm2a* cKO c-KIT-positive cells. (N) Venn diagram showing the overlap between *Kdm2a*-bound genes with increased H3K36me3 signal genes and *Kdm2a*-bound upregulated genes. (O) Venn diagram showing the overlap of *Kdm2a*-bound genes with increased H3K36me3 signal genes and *Kdm2a*-bound downregulated genes. Source data are available online for this figure.

As mentioned above, E2F1 usually works with its cofactor HCFC1 to control gene expression. However, we were surprised to find that their operation in mouse testes was abrogated upon *Kdm2a* deletion (Appendix Fig. S4A), indicating an essential role of KDM2A in forming the E2F1-HCFC1 complex. Next, we asked whether the deposition of E2F1 and HCFC1 on the genetic region was affected by KDM2A ablation. The CUT&RUN-qPCR results showed that the ability of both E2F1 and HCFC1 to bind the promoter of target genes was abolished due to KDM2A ablation in male germ cells (Fig. 7A,B; Appendix Fig. S4B,C). These results suggest that the E2F1-HCFC1 complex could be recruited to the transcriptional regulatory region of KDM2A target genes.

To investigate whether KDM2A can modulate gene transcription in collaboration with HCFC1 and E2F1, we selected four meiotic-related target genes (*Stra8*, *Meiosin*, *Spo11*, and *Sycp1*) that were down-regulated in *Kdm2a* cKO for luciferase reporter assays. We first transfected the luciferase reporter plasmids containing the target gene promoters 2 kb upstream of the transcription start site (TSS) together with the vectors overexpressing Flag-tagged *Kdm2a*, Myc-tagged *Hcfc1* or *E2f1*. The WB results confirmed the successful expression of these tagged proteins (Appendix Fig. S4D–F). In comparison to the control cells transfected only Flag- or Myc-tagged empty plasmids, the reporter activities were unchanged (*Meiosin*, *Sycp1*, and *Spo11*) or slightly increased (*Stra8*) in cells expressing KDM2A but significantly elevated in cells expressing E2F1 (Fig. 7C–F). Interestingly, when E2F1 was co-overexpressed with KDM2A and HCFC1, the reporter activities were markedly higher than any other group (Fig. 7C–F), indicating that the transcriptional activity of E2F1 for target genes strongly depended on the existence of KDM2A and HCFC1. Next, we performed knockdown experiments to further verify this. High knockdown efficiencies of designed siRNA targeting the mRNA of KDM2A, HCFC1, and E2F1 genes were validated by WB assay (Appendix Fig. S4G–I). Of note, the knocking down of KDM2A or E2F1 resulted in decreased reporter activities, but it can be partially rescued by the overexpression of E2F1. However, when we knocked down both KDM2A and HCFC1 and then overexpressed E2F1, the transcriptional activity of the target gene was unable to be rescued (Fig. 7G–J). These results highlight that KDM2A can work with HCFC1 and E2F1 to promote the transcription of these downregulated genes. To determine whether upregulated genes can also be regulated by KDM2A via cooperation with HCFC1 and E2F1, among which four genes (*Sox3*, *Trim71*, *Sall4*, and *Eomes*) that are associated with spermatogonia development were selected for the luciferase reporter

assay. The results showed that the expression of these four genes can be suppressed by KDM2A and E2F1, but not by HCFC1 (Appendix Fig. 4J–Q), suggesting that KDM2A and E2F1 may also be involved in regulating the transcription of these upregulated genes. Combining the various sequencing data, we found a large number of common peaks bound by E2F1 and HCFC1 that can overlap that of either H3K36me2 or H3K36me3 (Appendix Fig. S4R,S). Furthermore, almost half of up- or down-regulated genes with abnormal H3K36me3 signal also showed a decreased HCFC1 and E2F1 enrichment (Appendix Fig. S4T,U), suggesting a possible functional relevance between H3K36me2/3 modification and HCFC1 and E2F1 accumulated on coordinating the expression of genes essential for meiosis entry and progression. Taken together, these findings revealed that KDM2A could not only orchestrate the H3K36me2/3 modification but also control the recruitment of HCFC1 and E2F1 to regulate its targeting gene expression in male germ cells.

## Discussion

Several histone methylations contribute to the formation of a highly complicated and well-organized regulatory network during sperma-togenesis, ensuring that this process sequentially progresses according to a well-designed program and ultimately produces mature male gametes (Wang et al, 2017). Dynamic changes in histone methylation facilitate the regulation of transcriptional activity of genes participating in meiosis of male germ cells. In this study, we demonstrated that a histone demethylase, KDM2A, undergoes dynamic changes during spermatogenesis and plays an essential role in male meiosis. Conditional deletion of *Kdm2a* in germ cells using a newly developed *Stra8GFP-Cre* knock-in mouse line, in which CRE recombinase is expressed in both sexes (Lin et al, 2017), severely inhibited meiotic entry and caused defective meiotic progression. Interestingly, another study reported that conditional knockout of *Kdm2a* in male germ cells by earlier transgenic *Stra8-Cre* did not impact male fertility, although spermatogenesis was partially impaired (Xiong et al, 2023). Overall, our findings highlighted the crucial physiological roles of KDM2A in maintaining the balance of H3K36me2/3 deposition on promoters of functional genes in pre-meiotic germ cells. Furthermore, we revealed that the regulation of transcriptional activity of key genes partially depends on the cooperation of KDM2A with HCFC1 and E2F1, which expanded our understanding of the function of histone methylations on the regulation of meiosis entry and progression.

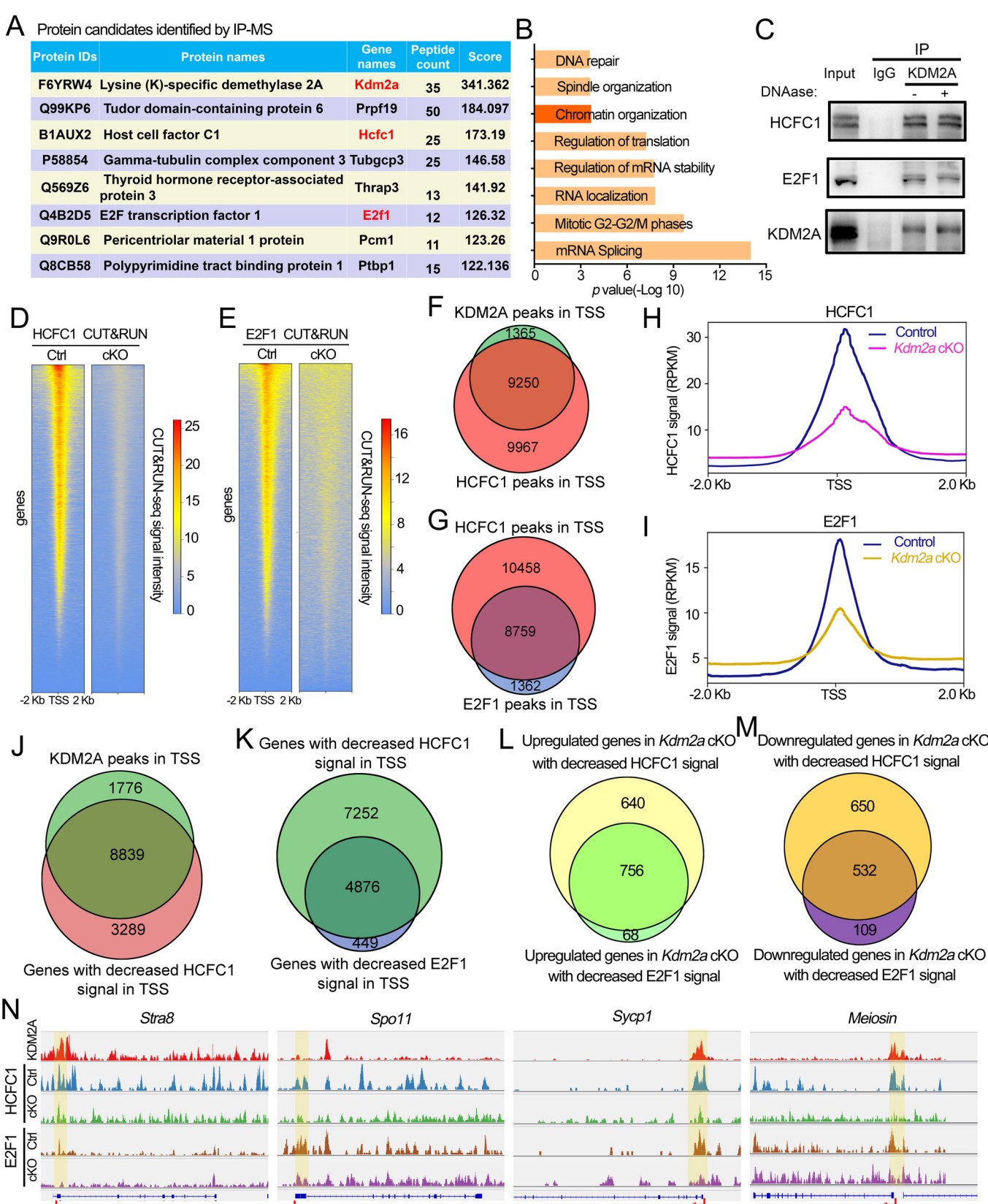

◄

**Figure 6. KDM2A interacts with HCFC1 and E2F1 to regulate gene expression in germ cells.**

(A) A list of eight KDM2A-interacting partners with highest score in sorted c-KIT⁺ cells identified by immunoprecipitation-mass spectrometry (IP-MS). (B) GO analyses of KDM2A-interacting proteins identified from IP-MS data. (C) Validation of interactions between KDM2A and two putative KDM2A-interacting proteins (HCFC1 and E2F1) in purified c-KIT-positive cells by co-IP assays. IgG was used as a negative control. (D, E) Heatmap of HCFC1 and E2F1 expression levels in control and *Kdm2a* cKO c-KIT-poistive cells. HCFC1 and E2F1 expression levels are shown on the genomic regions between −2.0 kb upstream and +2.0 kb downstream of TSS. (F) Venn diagram showing the overlap between *Kdm2a*-bound genes and the *Hcfc1*-bound genes in TSS region. (G) Venn diagram showing the overlap between the *Hcfc1*-bound genes and the *E2f1*-bound genes in TSS region. (H, I) Average HCFC1 (H) or E2F1 (I) CUT&RUN signals of *Kdm2a* cKO c-KIT-positive cells versus control c-KIT-positive cells within −2 kb/+2 kb of TSS. (J) Venn diagram shows an overlapping of *Kdm2a*-bound genes of TSS in WT c-KIT-positive cells and genes with decreased HCFC1 signal of TSS in *Kdm2a* cKO c-KIT-positive cells. (K) Venn diagram depicting the overlap of genes in TSS region with decreased HCFC1 and E2F1 signal occupancy in *Kdm2a* cKO c-KIT-positive cells. (L) Venn diagram showing the overlap between upregulated genes with decreased HCFC1 and E2F1 signal in *Kdm2a* cKO c-KIT-positive cells. (M) Venn diagram showing the overlap between downregulated genes with decreased HCFC1 and E2F1 signal in *Kdm2a* cKO c-KIT-positive cells. (N) Genome browser tracks depicting reads accumulation of HCFC1 and E2F1 on representative meiotic genes *Stra8, Spo11, Sycp1, and Meiosin* in control and *Kdm2a* cKO c-KIT-positive cells using IGV software. Source data are available online for this figure.

Histone H3 lysine 36 (H3K36) methylation is critical for the organization of chromatin into distinct domains and the regulation of various biological processes, including spermatogenesis (Sharda and Humphrey, 2022). Specifically, the nuclear SET domain (NSD)-containing methyltransferases (NSD1, NSD2, and NSD3) facilitated H3K36me1/2 formation, while SET domain-containing 2 (SETD2) could uniquely produce H3K36me3 (Li et al, 2019). NSD1 broadly deposited H3K36me2 in euchromatic regions and played an important role in initiating DNA methylation in prospermatogonia (Shirane et al, 2020); NSD2 was expressed in pachytene spermatocytes and round spermatids, and knockout of *Nsd2* resulted in significantly reduced levels of H3K36me2 and H3K36me3 in mouse testes (Li et al, 2022). Similarly, SETD2 was predominantly located in the nuclei of pachytene spermatocytes and round spermatids, and ablation of *Setd2* in mice resulted in a complete absence of H3K36me3 in germ cells (Zuo et al, 2018). Notably, although these H3K36 methyltransferases function as key regulators during spermatogenesis, it is unknown whether and how demethylase targeting of H3K36me1/2/3 is required for male fertility. KDM2A, a modular protein containing JmjC, CXXC-zinc finger (ZF-CXXC), plant homologous zinc finger (PHD), and F-box-like domains, was the first identified JmjC-containing histone lysine demethylase (Tsukada et al, 2006) and exhibited specificity for removal of methyl groups from lower methylation states of H3K36 (H3K36me1/2). In the current study, we revealed that loss of the demethylase KDM2A results in defective male meiosis and male infertility. In contrast to the NSD2 and SETD2, we found that KDM2A is mainly expressed in spermatogonia and early spermatocytes, thus establishing a specific expression profile for H3K36me2/3 in postnatal germ cells. Unexpectedly, we found that KDM2A may function as a demethylase targeting H3K36me3 in mouse testis because most of the KDM2A-bound genes showed enhanced H3K36me3 deposition accompanied by decreased H3K36me2 signal upon KDM2A deletion, which is the first confirmed that KDM2A can catalyze H3K36me3 under physiological conditions. A previous study revealed the ability of KDM2A to bind the H3K36me3 peptide; however, the steric constraints were supposed to prevent KDM2A from executing the demethylation reaction (Cheng et al, 2014). It was reported that the trimethylated lysine fits the catalytic pocket and is possibly stabilized through cation–π interaction, and such interaction occurs in many biological structures, including the PHD domain (Taverna et al, 2007). Thus, we speculated that the PHD domain of KDM2A may facilitate it to bind the H3K36me3 peptide and undergo a demethylation reaction with the help of an unknown factor to release the steric constraints in a special reaction pool provided by testicular germ cells. Nevertheless, it will be interesting to determine whether a potential allosteric effect on KDM2A activity exists in mouse testes.

Furthermore, histone methylation plays a critical role in maintaining the balance between self-renewal of spermatogonia and commitment to meiosis (Griswold, 2016). In type-B spermatogonia, transcriptionally activating modifications such as H3K4me3 gradually increased and induced the expression of genes that regulate cell differentiation and meiotic entry. Conversely, H3K27me3, linked with repression of transcriptional activity, favors the silencing of genes probably involved in self-renewal maintenance (Hammoud et al, 2014). Interestingly, our study found that KDM2A could control the deposition of H3K36me2/3 on the promoter of a number of genes involved in spermatogonial development and meiotic progression. In addition to providing signals for modulating transcription activity, H3K36me3 also contributes to pre-messenger RNA (mRNA) splicing by recruiting splicing-specific regulators, DNA damage repair by gathering the key mismatch sensor hMutSα onto chromatin, and DNA methylation by recruiting the DNA methyltransferases DNMT3A and DNMT3B (Weinberg et al, 2019). In fact, our IP-MS data identified some potential KDM2A interacting partners involved in 'mRNA splicing' and 'DNA repair', suggesting that a possible function of KDM2A on this regulatory mechanism during spermatogenesis cannot be excluded. Future work aimed at understanding how KDM2A-mediated regulation of H3K36me2/3 impacts and integrates with other histone methylation should reveal critical new insights into the role of chromatin in regulating meiosis entry. Of note, almost half of the KDM2A-targeted genes with a reduction in H3K36me2 were not accompanied by an increase of H3K36me3 (Appendix Fig. S3D). There are two reasons that could explain this phenomenon: (i) Although our data showed that *Kdm2a* deletion caused an increase in H3K36me2 and a decrease in H3K36me3 in germ cells, we cannot exclude the possibility that KDM2A may function as a demethylase targeting H3K36me2, just like its traditional role in somatic cells, which can bind some genes and catalyze H3K36me2 deposited in these genes to H3K36me1. (ii) In addition to KDM2A, other H3K36 methylases and demethylases may also play a role in germ cell development. It is worth noting that the other two H3K36me3 demethylases (Sharda and Humphrey, 2022), KDM4A and KDM4C, were found to be up- and down-regulated, respectively, in *Kdm2a* cKO germ cells (Dataset

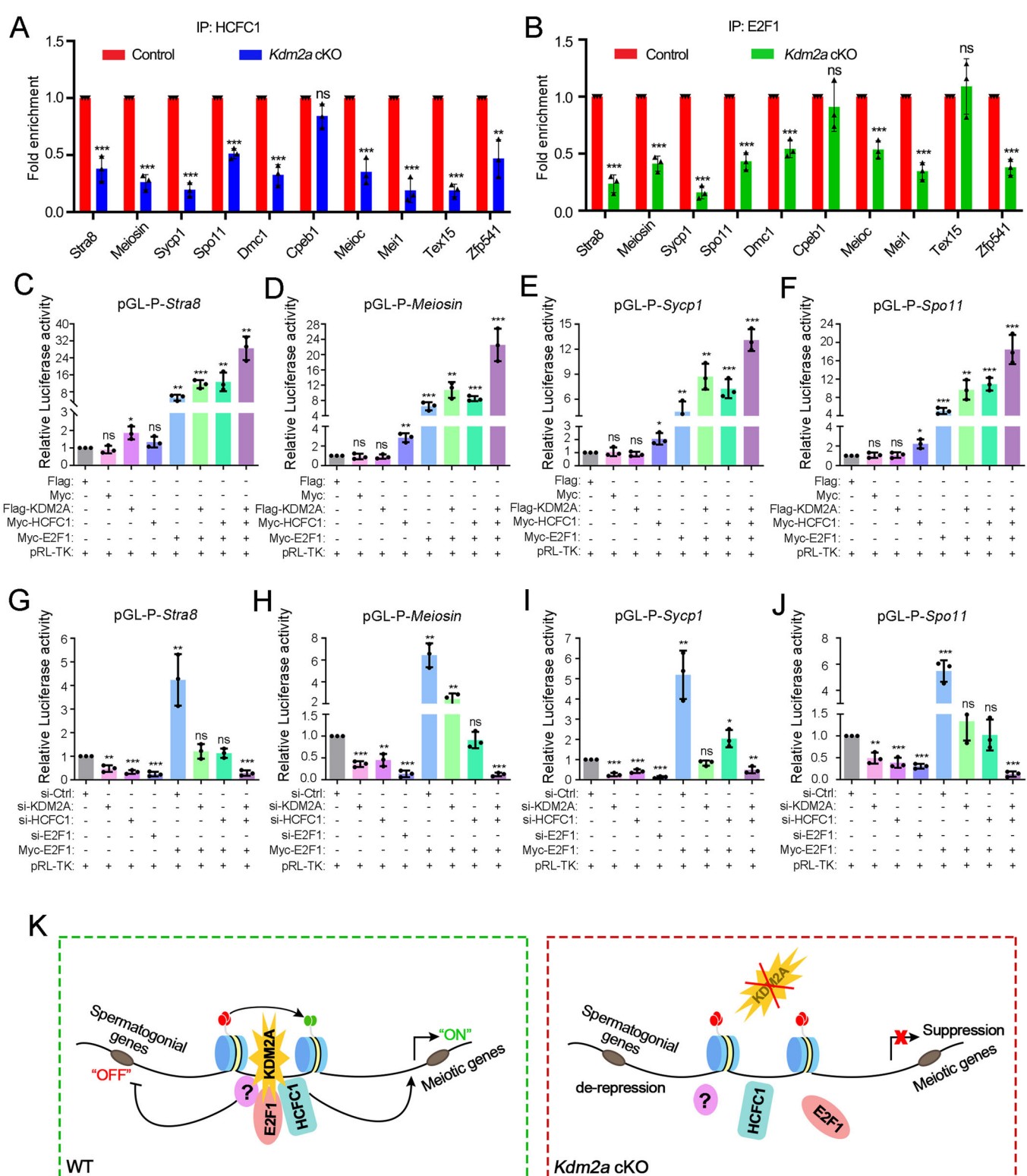

EV1), suggesting that they may compete or cooperate with each other to regulate the balance of H3K36me2 and H3K36me3 distribution on KDM2A-targeted genes. However, we cannot rule out the involvement of other factors in this process, which will be worthy of further investigation in the future.

In addition, one of the striking findings in the current study is that our data revealed a functional dependence of KDM2A on HCFC1 and E2F1 during meiotic entry in spermatogenesis and the binding of HCFC1 and E2F1 at targeted genomic loci is reduced in *Kdm2a* cKO c-KIT-positive cells. Our results suggested that KDM2A could guide

**Figure 7. KDM2A recruits HCFC1 and E2F1 on the promoter of target genes.**

(A, B) RIP-qPCR analyses of the association of the selected gene mRNAs with HCFC1 (A) and E2F1 (B) in control and *Kdm2a* cKO c-KIT-positive cells are shown. Biologically independent mice (*n* = 3) were examined in three separate experiments. Data are presented as mean ± SD. *P* value was calculated using a two-tailed Student's *t*-test. ns, not significant. For (A), left to right: \*\*\**P* = 0.0006, \*\*\**P* < 0.0001, \*\*\**P* < 0.0001, \*\*\**P* < 0.0001, \*\*\**P* = 0.0002, \*\*\**P* = 0.0005, \*\*\**P* = 0.0001, \*\*\**P* < 0.0001, \*\**P* = 0.0051; For (B), left to right: \*\*\**P* < 0.0001, \*\*\**P* = 0.0001, \*\*\**P* < 0.0001, \*\*\**P* = 0.0001, \*\*\**P* = 0.0005, \*\*\**P* = 0.0004, \*\*\**P* = 0.0001, \*\*\**P* = 0.0001. Source data are available online for the RIP-qPCR. (C–F) Luciferase reporter gene assays show that when *Kdm2a*, *Hcfc1*, and *E2f1* were overexpressed in different combinations, the luciferase activity of the *Stra8* (C), *Meiosin* (D), *Sycp1* (E), and *Spo11* (F) promoter regions in HEK293T cells was significantly increased. Data are presented as mean ± SD. *n* = 3 biological replicates. *P* value was calculated using a two-tailed Student's *t*-test. ns, not significant. For (C), left to right: \**P* = 0.0160, \*\**P* = 0.0060, \*\*\**P* = 0.0007, \*\**P* = 0.001; For (D), left to right: \*\**P* = 0.0021, \*\*\**P* = 0.0009, \*\**P* = 0.0012, \*\*\**P* < 0.0001, \*\*\**P* = 0.0009; For (E), left to right: \**P* = 0.0144, \*\**P* = 0.0080, \*\**P* = 0.0010, \*\*\**P* = 0.0007, \*\*\**P* < 0.0001; For (F), left to right: \**P* = 0.0103, \*\*\**P* = 0.0005, \*\**P* = 0.0022, \*\*\**P* = 0.0003, \*\*\**P* = 0.0007. (G–J) Luciferase reporter gene assays show that when *Kdm2a, Hcfc1,* and *E2f1* were knocked down in different combinations, the luciferase activity of the *Stra8* (G), *Meiosin* (H), *Sycp1* (I), and *Spo11* (J) promoter regions in HEK293T cells was remarkable decreased. Data were presented as mean ± SD. *n* = 3 biological replicates. *P* value was calculated using a two-tailed Student's *t*-test. ns, not significant. For (G), left to right: \*\**P* = 0.0030, \*\*\**P* = 0.0001, \*\*\**P* = 0.0002, \*\**P* = 0.0068, \*\*\**P* = 0.0004; For (H), left to right: \*\*\**P* < 0.0001, \*\**P* = 0.0025, \*\*\**P* < 0.0001, \*\**P* = 0.0010, \*\**P* = 0.0094, \*\*\**P* < 0.0001; For (I), left to right: \*\*\**P* = 0.0001, \*\*\**P* = 0.0004, \*\*\**P* < 0.0001, \*\**P* = 0.0037, \**P* = 0.0140, \*\**P* = 0.0063; For (J), left to right: \*\**P* < 0.0023, \*\*\**P* = 0.0008, \*\*\**P* < 0.0001, \*\*\**P* = 0.0007, \*\*\**P* < 0.0001. (K) Schematic working model of the KDM2A, HCFC1, and E2F1 complexes that regulate the completion of the meiotic prophase I. Source data are available online for this figure.

HCFC1 and E2F1 in targeting specific genomic loci to regulate gene expression through consensus sequences. Of note, the binding of HCFC1 or E2F1 at some genomic loci was not completely abolished without KDM2A, suggesting that other transcription factors may also be involved. Moreover, conditional knockout of KDM2A in male germ cells suppressed meiotic genes and elevated spermatogonial genes at mitosis-to-meiosis transition stages, indicating that KDM2A can be both a coactivator and corepressor. Considering both KDM2A and its interacting protein E2F1 could directly bind to many upregulated and downregulated genes, other potential factors should exist to distinguish the different mechanisms by which KDM2A exerts its inhibitory or active function on gene expression.

Interestingly, it has been reported that STRA8 is able to bind and enormously enhance the expression of E2F1 (Kojima et al, 2019). Moreover, at the TSS of STRA8-activated genes, the most enriched known motifs were those for E2F transcription factors, suggesting that the genes are regulated in a cell cycle-dependent manner. In the current study, we demonstrated that E2F1 and its co-factor HCFC1 can be recruited by KDM2A to the promoter of *Stra8* and *Meiosin* and facilitate their expression. A recent study revealed an essential role of the histone variant H3t in spermatogonial differentiation and meiosis entry (Ueda et al, 2017). Notably, the expression of H3t is closely regulated by *Stra8* and *Meiosin* in addition to KDM2A (Ishiguro et al, 2020). Furthermore, *Znhit1*-dependent H2A.Z deposition is required for *Meiosin* transcription, which is essential for meiotic initiation (Sun et al, 2022). Coincidentally, E2F1 was identified as one downstream target of H2A.Z (Xiang et al, 2023), and conversely, the H2A.Z gene transcription can be activated by E2F1 (Yoon et al, 2022). Therefore, our findings combined with these studies suggest that a complex interconnected network of synergistic activation surrounds the initial entry of germ cells into meiosis.

As meiosis is a unique feature of the germ cell lineage, insights into how histone modification and other factors interact to achieve an appropriate balance between proliferation and differentiation during the mitosis-to-meiosis transition will further our understanding of the molecular mechanisms underlying meiosis entry and progression. Our study uncovers an essential role for KDM2A in orchestrating the H3K36me2/3 modification and recruiting the key transcription factor E2F1 and the cofactor HCFC1 to the promoters of its target genes, and establishes a model whereby KDM2A controls a network of programmed gene expression in

male germ cells to ensure the transition from mitosis to meiosis (Fig. 7K). The present study adds to our understanding of histone demethylase and transcription factor cooperation in the transcriptional regulation of genes essential for meiosis entry and progression. However, studies are needed to investigate whether KDM2A-mediated H3K36me2/3 deposition is required to recruit E2F1 and HCFC1 to KDM2A target genes and further regulate gene expression during meiosis and germ cell development.

## Methods

### Mice

The *Kdm2a*^flox mouse line was generated as described previously. For germ cell-specific knockout of *Kdm2a*, the *Stra8-GFPCre* knock-in mouse line was used (Lin et al, 2017). The *Stra8-GFPCre* mouse line in the C57BL/6J background was obtained from Dr. Minghan Tong's Laboratory at the Center for Excellence in Molecular Cell Science, Chinese Academy of Sciences. Six-week-old *Stra8-GFPCre* males were first crossed with 6-week-old *Kdm2a*^flox/flox females to generate the *Stra8-GFP*Cre; *Kdm2a*^+/flox males, then the 8-week-old *Stra8-GFPCre*; *Kdm2a*^+/flox male mice were bred with *Kdm2a*^flox/flox female mice to obtain the *Stra8-GFP*Cre; *Kdm2a*^flox/Del (designated as *Kdm2a* cKO) male mice. All mice in this study were maintained on C57BL/6J strains and housed in specific pathogen-free conditions at the Laboratory Animal Center of Huazhong University of Science and Technology. All animal experiments were approved by the Institutional Animal Care and Use Committee (IACUC) of Tongji Medical College, Huazhong University of Science and Technology. Tamoxifen injections were performed as previously described (Wen et al, 2024). Briefly, the *DDX4-Cre*^ERT2; *Kdm2a*^flox/flox male mice and their littermate controls were treated by intraperitoneal injection of Tamoxifen (100 μl at 20 mg/mL in corn oil) for 5 consecutive days. Matched male littermates aged 8 weeks were used in this experiment. Histological analysis was performed 35 days after tamoxifen injection.

### Fertility test

Adult control and *Kdm2a* cKO male mice were bred with adult wild-type (WT) C57BL/6J female mice, which had been confirmed to be fertile, in a ratio of 1:2. Successful sexual intercourse was defined by the

presence of vaginal plug. The plugged females were kept separately in cages every morning. The pregnancy and the litter sizes were recorded. The fertility testing was lasting for at least 6 months.

## Histological analysis

After euthanasia, the testes and epididymides from control and *Kdm2a* cKO mice were collected and fixed in Bouin's solution (Sigma, HT10132) overnight at 4 °C, then embedded in paraffin and sectioned. The slides were stained with periodic acid-Schiff (PAS) using a standard protocol and imaged with a light microscope (Axio Scope.A1, Zeiss, Germany).

## Quantitative real-time PCR (RT-qPCR)

Total RNA was extracted using TRIzol reagent (Invitrogen) according to the manufacturer's instructions. cDNA synthesis was carried out using the PrimeScript RT reagent Kit with gDNA Eraser (TaKaRa, RR047B). RT-qPCR analysis was performed using SYBR® green master mix (YEASEN) and Step One ABI real-time PCR System. Primer sequences were listed in Appendix Table S1.

## Immunofluorescence (IF)

The mouse testes were collected and fixed in 4% paraformaldehyde (PFA) in PBS overnight at 4 °C. The samples were embedded in a Tissue-Tek optimal cutting temperature (O.C.T, Sakura Finetek, 4583) compound after dehydration sequentially. Then, the 5 μm cryosections were prepared. The slides were boiled in 10 mM sodium citrate buffer (pH = 6.0) for 15 min using a microwave oven for antigen retrieval. After washing using PBS, sections were blocked in 5% BSA for 1.5 h at room temperature (RT), and incubated with primary antibodies at 4 °C overnight. After washing, the slides were incubated with a secondary antibody protecting from light for 1 h at RT. Slides were counterstained with DAPI (H1200, Vector laboratories). Images were captured using a Laser confocal scanning microscopy. All antibodies used in this study were listed in Appendix Table S2.

## Chromosome spreading assay

Chromosome spreads were prepared as previously reported with slight modifications (Feng et al, 2022). In brief, testicular seminiferous tubules were pretreated by hypotonic buffer (30 mM Tris, 50 mM sucrose, 17 mM trisodium citrate dihydrate, 5 mM EDTA, 0.5 mM DTT, and 0.5 mM phenylmethylsulphonyl fluoride (PMSF), pH = 8.2) for 60 min. The tubules were then cut into several short fragments and suspended in 100 mM sucrose to disperse to a single cell and spread to a thin cell layer on slides. Followed with 2% (w/v) paraformaldehyde solution containing 0.15% Triton X-100 treatment, the prepared slides were placed in a humidified chamber for 4 h at RT and washed with 0.4% Photo-Flo (Kodak, 1464510). The slides were then air-dried at RT for 15–30 min and then blocked with blocking solution (containing 5% normal donkey serum) for 1 h for immunostaining.

## TUNEL assay

Detection of apoptotic cells was performed using the One Step TUNEL Apoptosis Assay Kit (Meilunbio, MA0223) according to

the manufacturer's instructions. DAPI was used to label the nucleus.

## Whole-mount staining

Adult mouse seminiferous tubules whole-mount immunofluorescence was performed as previously reported with slight modification (Wang et al, 2019). Briefly, the testis was isolated, and the seminiferous tubules were disentangled and then immediately fixed in 4% PFA at 4 °C overnight. After fixation, the tubules were dehydrated in a series of graded methanol dilutions, then permeabilized with 0.5% Triton X-100 in PBS and blocked with 1% BSA and 5% donkey serum in PBS at RT for 2 h. Following three times washes (10 min per wash), the tubules were incubated with the indicated primary antibody overnight at 4 °C. After washing, seminiferous tubules were incubated 2 h at RT with species-specific secondary antibodies conjugated to Alexa488 or Cy3 fluorochromes. Tubules were mounted on slides with raised coverslips for fluorescence imaging. The nuclei were stained with DAPI.

## Western blotting analysis

Protein from the indicated samples was extracted using RIPA lysis buffer (CWBIO, Cat# CW2333S) with the addition of a protease inhibitor cocktail (P1010, Beyotime). The proteins were then separated on a 10% SDS-PAGE gel and transferred to the PVDF membranes (Bio-Rad). The membranes were blocked with 5% skimmed milk for 1 h at RT, followed by incubation with primary antibodies overnight at 4 °C. On the second day, after washing with TBST, the membranes were incubated with secondary antibodies at RT for 2 h and photographed using the Luminol/enhancer solution and Peroxide solution (ClarityTM Western ECL Substrate, Bio-Rad) and the ChemiDoc XRS+ system (Bio-Rad).

## Purification of spermatogonia cells by MACS

c-KIT+ spermatogonial cells were isolated using CD117 MicroBeads (Miltenyi, 130-091-224) according to the manufacturer's instructions. In brief, freshly harvested testes from P10 mice were washed with ice-cold PBS three times, then decapsulated and digested using collagenase IV (1 mg/ml) at 37 °C for 10 min. After allowing the seminiferous tubules to settle and discarding the supernatant, the cells washed twice with ice-cold PBS and incubated in trypsin (0.05%) and DNase I (0.5 mg/ml) for 2–3 min at 37 °C, then the digestion was terminated, and the cell suspensions were filtered (40 μm). MACS buffer and CD117 MicroBeads were then added to the filtered cells and incubated for 15 min at 4 °C. Finally, c-KIT+ spermatogonia cells were purified using a magnetic column and stored at −80 °C for subsequent experiments.

## Purification of testicular germ cells

To isolate testicular germ cells from the control and *Kdm2a* cKO juvenile mice (P10), testes were harvested and dissociated into single cells according to established protocols. Testis samples were digested with collagenase IV (1 mg/ml), followed by washing with DMEM and centrifugation at $400 \times g$ for 5 min. The pellet was then subjected to a second digestion with 0.25% trypsin with

DNase I to obtain a single-cell suspension. The resulting suspension was then plated onto 0.1% gelatin-coated plates and incubated for 2 h. During this incubation, somatic cells adhered to the culture plates while germ cells remained non-attached. Germ cells from control or *Kdm2a* cKO testes were enriched by gelatin selection by collecting the non-attached cells. Fractions containing germ cells were then centrifuged and collected for subsequent procedures.

## Cell culture, transfection, and luciferase reporter assays

HEK293T cells were obtained from the Stem Cell Bank of Chinese Academic Science (Cat# GNHu43), and cultured in the DMEM medium supplemented with 10% fetal bovine serum (Gibco, 10270106). The full-length of Kdm2a cDNA was cloned into a pCMV vector containing the N-terminal Flag epitope tag, and the full-length Hcfc1 and E2f1 cDNAs were cloned into a pCMV vector containing the N-terminal c-Myc epitope tag. Luciferase reporter plasmids (pGL4.10) were linked to the *Stra8, Meiosin, Sycp1, Spo11, Sox3, Trim71, Sall4*, and *Eomes* promoters, respectively. Transfection of these plasmids was performed using Lipofectamine 2000 (Life Technologies) according to the manufacturer's instructions. For RNA interference, cells were transfected with appropriate siRNAs using Lipofectamine RNAiMAX (Invitrogen) and harvested 48 h later for analysis. The non-targeting siRNA, targeting human KDM2A siRNA, HCFC1 siRNA and E2F1 siRNA were purchased from (Sangon Biotech, China). For luciferase reporter assays, HEK293 cells were co-transfected with luciferase reporter plasmids (200 ng) and overexpression plasmid and/or siRNA when cells were ~80% confluent. Cells were then harvested and lysed 24 h later. Luciferase activity was measured using a dual-luciferase reporter assay system following the manufacturer's instructions (Promega). The primers for plasmid construction and siRNA sequence were listed in Appendix Table S1.

## Immunoprecipitation (IP) and mass spectrometry (MS)

For immunoprecipitation, c-KIT$^+$ spermatogonia were homogenized in ice-cold cell lysis buffer (Beyotime, P0013) containing protease inhibitor cocktail (P1010, Beyotime). Cell lysates were then incubated with relevant antibodies and pre-cleaned magnetic protein A/G beads (Bio-Rad, 161-4013), with or without Super Nuclease (Beyotime, D7121) overnight at 4 °C. The supernatants were then discarded and the beads were washed six times with ice-cold cell lysis buffer. Then 5× SDS loading buffer was added, followed by boiling for 10 min, and subjected to western blotting procedures.

For the identification of interaction proteins by MS, IP was performed as above. The washed beads were incubated with 40 μl elution buffer [10 mM EDTA, 50 mM Tris (pH 8.0), 1× protease inhibitor] at 70 °C for 30 min. The extracted proteins were analyzed on an analytical capillary column (50 μm × 10 cm), followed by spraying into an LTQ ORBITRAP Velos mass spectrometer (Thermo Fisher Scientific, San Jose, CA, USA) equipped with a nano-ESI ion source. The identified peptides were then searched in the IPI (International Protein Index) mouse protein database on the Mascot server (Matrix Science Ltd, UK).

## RNA-seq and data analysis

Total RNA was purified from c-KIT$^+$ spermatogonia of P10 WT and *Kdm2a* cKO mice using TRIzol reagents (Invitrogen). mRNA libraries were prepared using the NEBNext Ultra RNA Library Prep Kit for Illumina (New England Biolabs) according to the manufacturer's instructions and sequenced on the Illumina Hi-Seq 2500 platform, with three biological replicates for control and *Kdm2a* cKO mice.

Cutadapt v1.9.1 was used to process raw reads to remove adapters and perform quality trimming, and HiSAT2 (V2.0.1) was used to map the trimmed reads to UCSC mm10 components with default parameters. Differentially expressed genes in pairwise comparisons were measured by DESeq2 (v1.10.1), and the reads were internally normalized to account for library size and RNA composition bias. Differentially regulated genes in DESeq2 analysis were defined as fold-change >1.5 and *P*-value < 0.05. Gene Ontology (GO) analysis was performed using the DAVID database.

## CUT&RUN assay and analysis

CUT&RUN was performed using MACS-purified spermatogonia (c-KIT$^+$ cells) according to the previously described with minor modifications (Dura et al, 2022). Briefly, c-KIT$^+$ cells were centrifuged at 800 × g for 5 min and washed twice with 1.5 ml wash buffer. During the washing, ConA Beads Pro (10 μl per condition) were washed twice in 1.5 ml binding buffer and resuspended in 10 μl binding buffer per condition. c-KIT$^+$ cells were mixed with beads and rotated for 10 min at RT, and samples were divided into equal parts as required. Samples were then collected on magnetic beads and resuspended in 100 μl ice-cold antibody buffer containing one of the following antibodies: anti-KDM2A, anti-H3K36me2, and anti-H3K36me3. The samples were then incubated with pG-MNase enzyme for 1 h at 4 °C, followed by two washes with 1 ml digitonin buffer. Samples were resuspended in 100 μl of digitonin buffer and cooled to 4 °C before the addition of CaCl$_2$. Targeted digestion was performed for 60 min on ice, followed by the addition of 100 μl of 2 × Stop buffer. Samples were then incubated at 37 °C for 20 min to release cleaved chromatin fragments, centrifuged at 14,000 × g for 5 min at 4 °C and collected on a magnet. The supernatants containing the cleaved chromatin fragments were then transferred and cleaned up using the Zymo Clean & Concentrator Kit. The CUT&RUN library was constructed according to the manufacturer's instructions and sequenced on a NovaSeq (Illumina) using PE 100 bp run, with three biological replicates each.

Paired-end reads were trimmed using Trim Galore v0.4.4 and aligned to the mouse genome (GRCm38) using Bowtie2 with the following settings: -local-very-sensitive-local-no-unal--q-phred33. Picard v2.6.0 was used to remove PCR duplicates, and MACS2 v2.1.1 was used to call peaks using the input sample as a control. A heatmap using peak centers as windows was generated using Deeptools v2.5.3. Motifs were identified using HOMER v4.11 with the mouse motif database as known motifs. Two-sample Mann–Whitney tests between the CPMs were used to test for differences in histone mark deposition between control and *Kdm2a* cKO samples.

## Single-cell RNA-seq data analysis

Raw single-cell RNA sequencing data were downloaded from the GEO database and pre-processed to filter out low-quality cells and reads, align the reads to the reference genome, and generate a count matrix representing gene expression levels in each cell. The normalized count matrix was then subjected to remove low-quality cells and genes with Seurat package. Dimensionality reduction and visualization techniques such as t-SNE are used to reduce the high-dimensional data to 2D for clustering and cell type identification. Marker gene expression patterns and reference databases were used to assign known cell types to different clusters. Trajectory analysis was also performed to infer lineage relationships using Monocle package. All analyses were performed in the R environment.

## Statistical analysis

Data in the present study are presented as mean ± standard deviation (SD). GraphPad Prism 9.0 software (GraphPad, San Diego, CA, USA) was used for the statistical analyses. Significant differences between the two groups were analyzed using the two-sided Student's *t*-test and two-tailed Mann–Whitney U-test, and differences between multiple groups were measured using one-way analysis of variance followed by Bonferroni post hoc tests. A value of $p < 0.05$ was considered statistically significant for any differences. *P*-values are denoted in figures or figure legends by $*p < 0.05$, $**p < 0.01$, and $***p < 0.001$.

# Data availability

All raw sequencing data are deposited in the NCBI SRA database with the accession number PRJNA1066023. The authors declare that all data supporting the findings of this study are available within the article and its appendix files or from the corresponding author upon reasonable request.

The source data of this paper are collected in the following database record: biostudies:S-SCDT-10_1038-S44318-024-00203-4.

# Peer review information

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

## Acknowledgements

We would like to thank Dr. Xianrong Xiong at Southwest Minzu University for kindly sharing the *Kdma2^flox* mice and Dr. Minghan Tong at Shanghai Institute of Biochemistry and Cell Biology, Chinese Academy of Sciences, for kindly sharing the *Stra8-GFPCre* mouse line. We also acknowledge the funding from the National Natural Science Foundation of China (82171605 and 82371625 to SY, 82101738 to JD, and 82071709 to XJ), the Joint Fund for Medical Artificial Intelligence (MAI2022Q010 to XJ), the Key Research and Development Project of Anhui Province (2022e07020014 to XJ), the Hubei Provincial Laboratory Animal Research Special Fund (2023CFA006 to SY), and the Basic Research Support Program of Huazhong University of Science and Technology (2023BR031 to SY).

## Author contributions

**Shenglei Feng**: Conceptualization; Data curation; Formal analysis; Validation; Investigation; Methodology; Writing—original draft; Project administration. **Yiqian Gui**: Conceptualization; Data curation; Formal analysis; Validation; Investigation; Methodology. **Shi Yin**: Conceptualization; Resources; Investigation. **Xinxin Xiong**: Investigation. **Kuan Liu**: Investigation; Methodology. **Jinmei Li**: Investigation. **Juan Dong**: Funding acquisition; Methodology. **Xixiang Ma**: Investigation. **Shunchang Zhou**: Investigation. **Bingqian Zhang**: Investigation. **Shiyu Yang**: Investigation. **Fengli Wang**: Project administration. **Xiaoli Wang**: Project administration. **Xiaohua Jiang**: Conceptualization; Resources; Funding acquisition. **Shuiqiao Yuan**: Conceptualization; Supervision; Funding acquisition; Investigation; Project administration.

Source data underlying figure panels in this paper may have individual authorship assigned. Where available, figure panel/source data authorship is listed in the following database record: biostudies:S-SCDT-10_1038-S44318-024-00203-4.

## Disclosure and competing interests statement

The authors declare no competing interests.

# Expanded View Figures

**Figure EV1.  The expression pattern of KDM2A and H3K36me1/2/3 during spermatogenesis.**

(**A**) RT-qPCR analyses of *Kdm2a* mRNA levels in various organs from wild-type (WT) adult mice. $n = 3$ (three biological replicates). Data are presented as the mean ± SD. (**B**) WB analyses the expression of KDM2A protein levels in multiple organs from WT adult mice. GAPDH was used as a loading control. Biologically independent mice ($n = 3$) performed three independent experiments. Data are presented as the mean ± SD. (**C**) RT-qPCR analyses of *Kdm2a* mRNA levels in WT testes at different developmental stages including postnatal day 0 (P0), P7, P14, P21, P28, P35, and adult. $n = 3$ (three biological replicates). (**D**) WB analyses the KDM2A protein levels in developing WT testes. GAPDH served as a loading control. Biologically independent mice ($n = 3$) performed three separate experiments. (**E**) Whole-mount staining of seminiferous tubules from adult WT testis with antibodies against KDM2A and PLZF. Nuclei were stained with DAPI. Scale bars = 50 μm. Examples of $A_{single}$ (As), $A_{paired}$ (Apr), and $A_{aligned}$ (Aal(4), Aal(8)) undifferentiated spermatogonia in adult testes (dotted circles). (**F**) same as (**E**) for adult WT testes, but with antibodies against KDM2A and c-KIT. Nuclei were stained with DAPI. Scale bars = 50 μm. (**G**) Co-Immunofluorescent staining of KDM2A and PLZF on testis sections from WT mice at P14. Nuclei were stained with DAPI. Scale bars = 50 μm. (**H**) Co-Immunofluorescent staining of SALL4 with H3K36me1 (upper), H3K36me2 (middle), and H3K36me3 (lower) on testis sections from WT mice at P14, respectively. Nuclei were stained with DAPI. Scale bars = 50 μm. The white arrows indicate the SALL4$^+$ spermatogonia. (**I**) Co-Immunofluorescent staining of SYCP3 with H3K36me1 (upper), H3K36me2 (middle), and H3K36me3 (lower) on surface-spread spermatocytes from WT mice at P21, respectively. Nuclei were stained with DAPI. Leptotene (Lep), Zygotene (Zyg), Pachytene (Pac), and Diplotene (Dip) spermatocytes are shown. Scale bars = 5 μm. (**J–L**) Co-Immunofluorescent staining of PLZF with H3K36me1 (**J**), H3K36me2 (**K**), and H3K36me3 (**L**) on whole-mount testes seminiferous tubules from adult WT mice. Nuclei were stained with DAPI. Scale bars = 50 μm. Source data are available online for this figure.

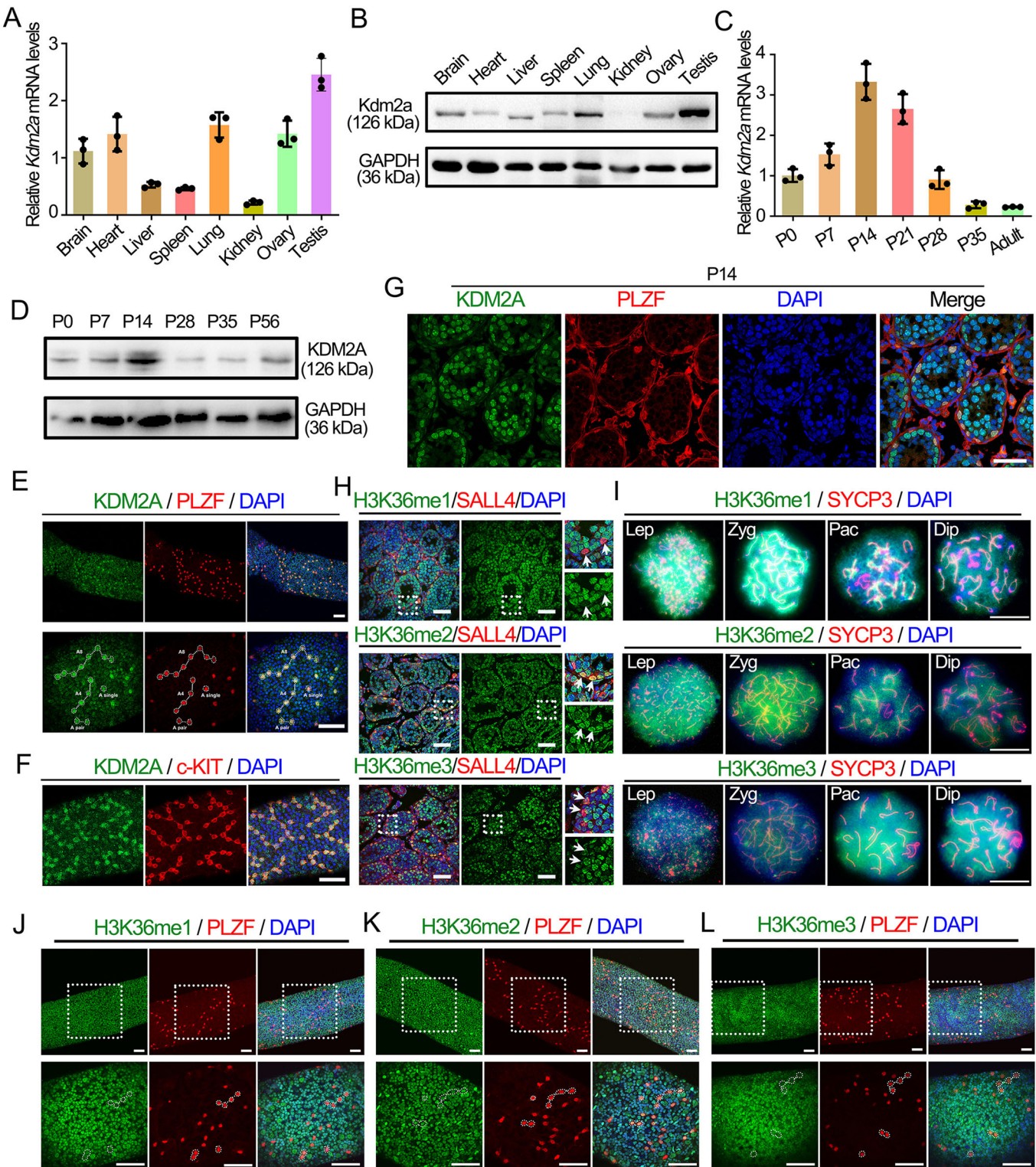

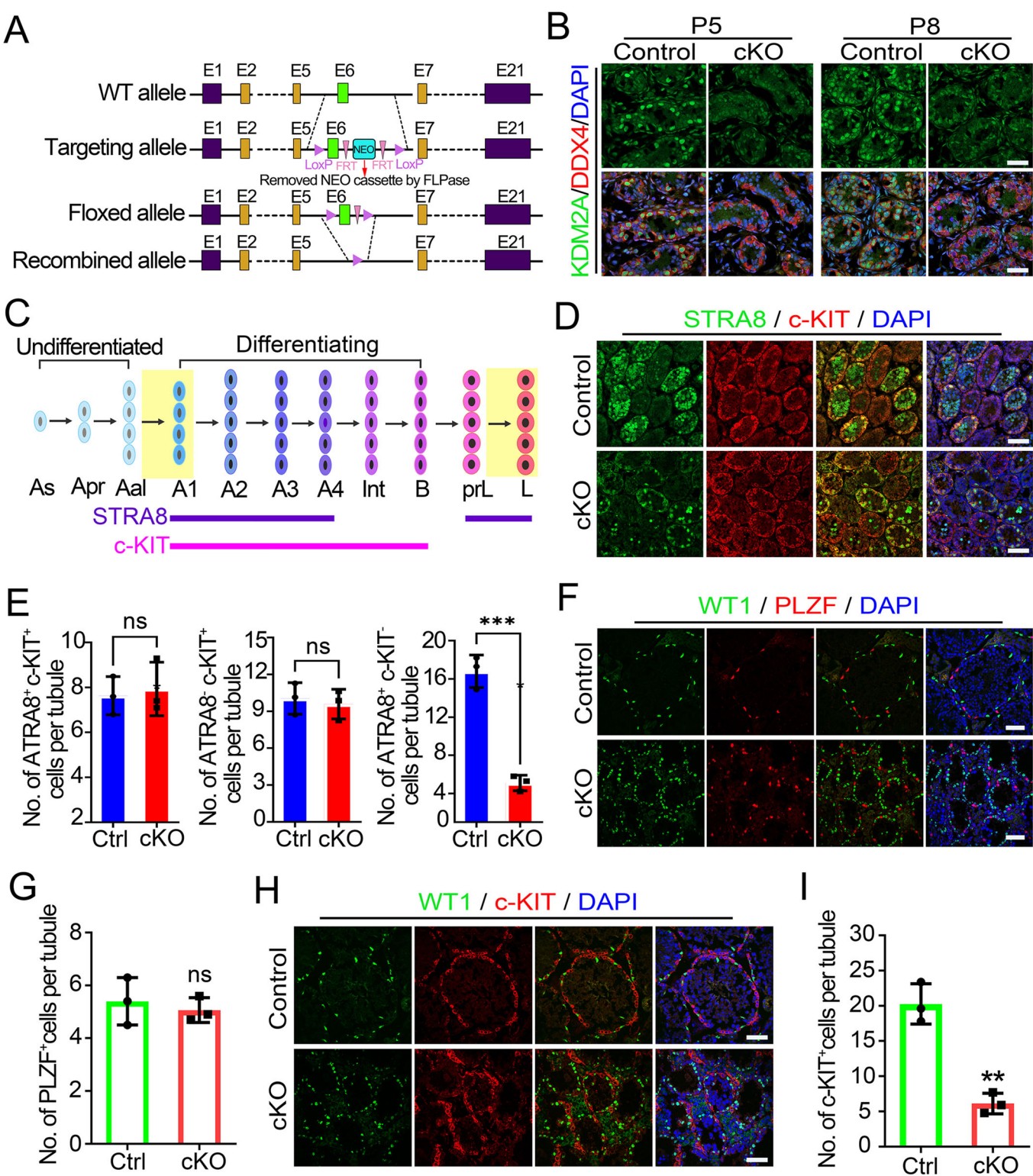

◄  **Figure EV2.   *Kdm2a* deletion results in defective differentiation of adult spermatogonia.**

(A) Schematic diagram of the targeting strategy used to generate floxed *Kdm2a* allele by homologous recombination in mouse embryonic stem cells.
(B) Immunofluorescence (IF) staining of KDM2A and DDX4 in control and *Kdm2a* cKO testes of P5 and P8 mice. Scale bars = 50 μm. (C) The Schematic diagram of spermatogonial differentiation process. As, A-single; Apr, A-paired; Aal, A-aligned; Int, intermediary spermatogonia; B, B spermatogonia; prL, preleptotene spermatocytes; and L, leptotene spermatocytes. (D) Co-immunofluorescent staining of STRA8 with c-KIT on testis sections from P10 control and *Kdm2a* cKO mice. Nuclei were stained with DAPI. Scale bars = 50 μm. (E) The quantifications of STRA8⁺ c-KIT⁺ or STRA8⁻ c-KIT⁺ or STRA8⁺ c-KIT⁻ cells per tubule. Data were presented as the mean ± SD. $n = 3$ biological replicates. *P* value was calculated using a two-tailed Student's *t*-test. ns, not significant. ***$P = 0.0004$. (F) Co-immunofluorescent staining of WT1 with PLZF on testis sections from adult control and *Kdm2a* cKO mice. Nuclei were stained with DAPI. Scale bars = 50 μm. (G) Quantification of PLZF⁺ cells per tubule for (F). Data were presented as the mean ± SD. $n = 3$ biological replicates. *P* value was calculated using a two-tailed Student's *t*-test. ns, not significant. (H) Co-immunofluorescent staining of WT1 with c-KIT on testis sections from adult control and *Kdm2a* cKO mice. Nuclei were stained with DAPI. Scale bars = 50 μm. (I) Quantification of c-Kit⁺ cells per tubule for (H). Data were presented as the mean ± SD. $n = 3$ biological replicates. *P* value was calculated using a two-tailed Student's *t*-test. ns, not significant. **$P = 0.0016$. Source data are available online for this figure.

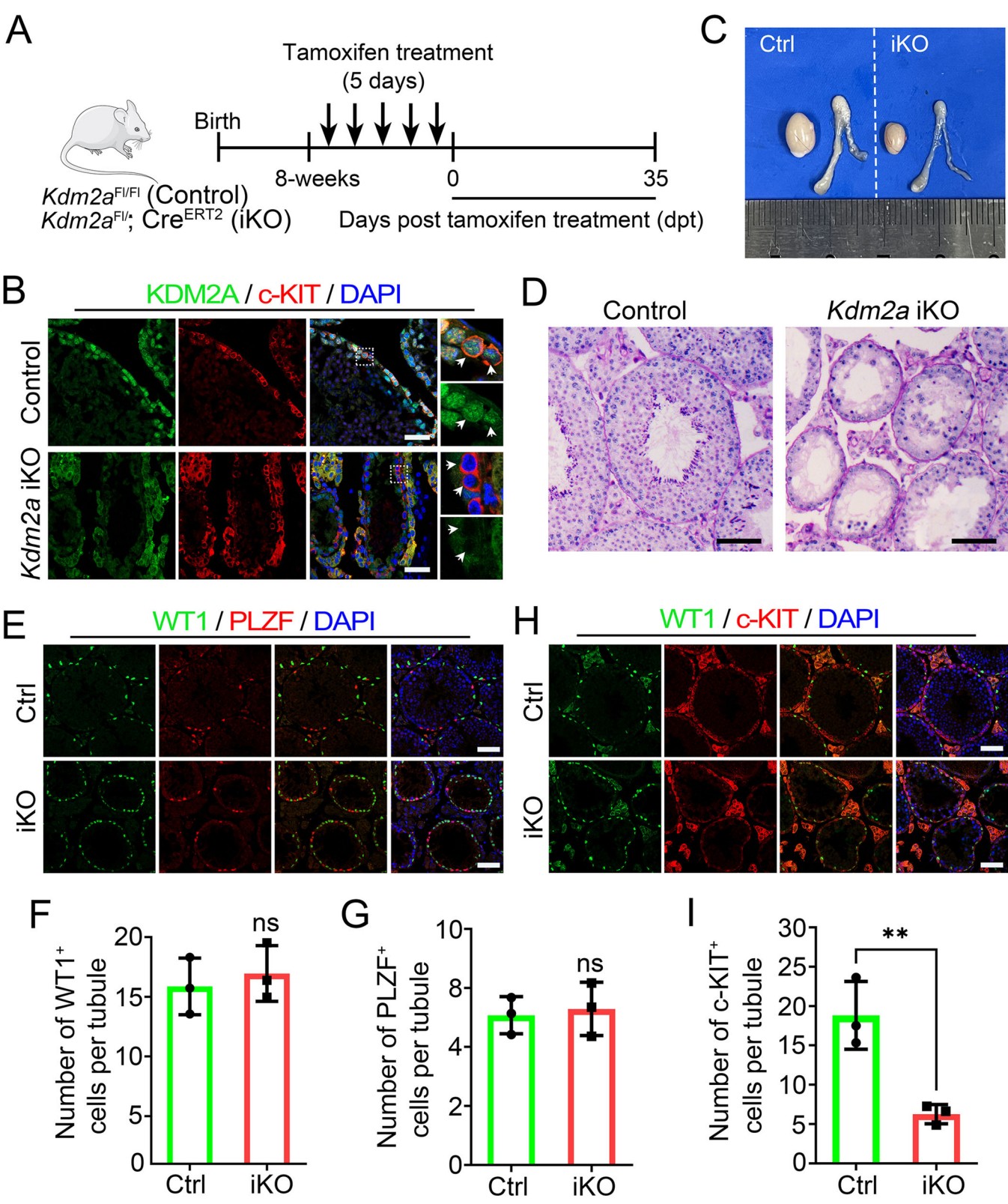

**Figure EV3. Tamoxifen-induced *Kdm2a* deletion in adult mice affects spermatogonial differentiation.**

(A) Regimen of tamoxifen treatment in adult (2 M) *Kdm2a*^fl/− Ddx4-Cre^ERT2 male mice. (B) Immunofluorescence (IF) staining of KDM2A and c-KIT in control and *Kdm2a* iKO testes. Arrowheads indicate KDM2A signals in the c-KIT positive cells. Nuclei were stained with DAPI. Enlarged images are shown in right panels. Scale bars = 50 μm. (C) Gross morphology of testes and epididymides from control and *Kdm2a* iKO mice. (D) Periodic acid-Schiff (PAS) staining of testes from control and *Kdm2a*-iKO mice. Scale bars = 50 μm. (E) Co-immunofluorescent staining of WT1 with PLZF on testis sections from adult control and *Kdm2a* iKO mice. Nuclei were stained with DAPI. Scale bars = 50 μm. (F, G) Quantification of WT1⁺ and PLZF⁺ (G) cells per tubule for (E). Data were presented as the mean ± SD. $n = 3$ biological replicates. *P* value was calculated using a two-tailed Student's *t*-test. ns, not significant. (H) Co-immunofluorescent staining of WT1 with c-KIT on testis sections from adult control and *Kdm2a* iKO mice. Nuclei were stained with DAPI. Scale bars = 50 μm. (I) Quantification of c-Kit⁺ cells per tubule for (H). Data were presented as the mean ± SD. $n = 3$ biological replicates. *P* value was calculated using a two-tailed Student's *t*-test. ns, not significant. **$P = 0.0083$. Source data are available online for this figure.

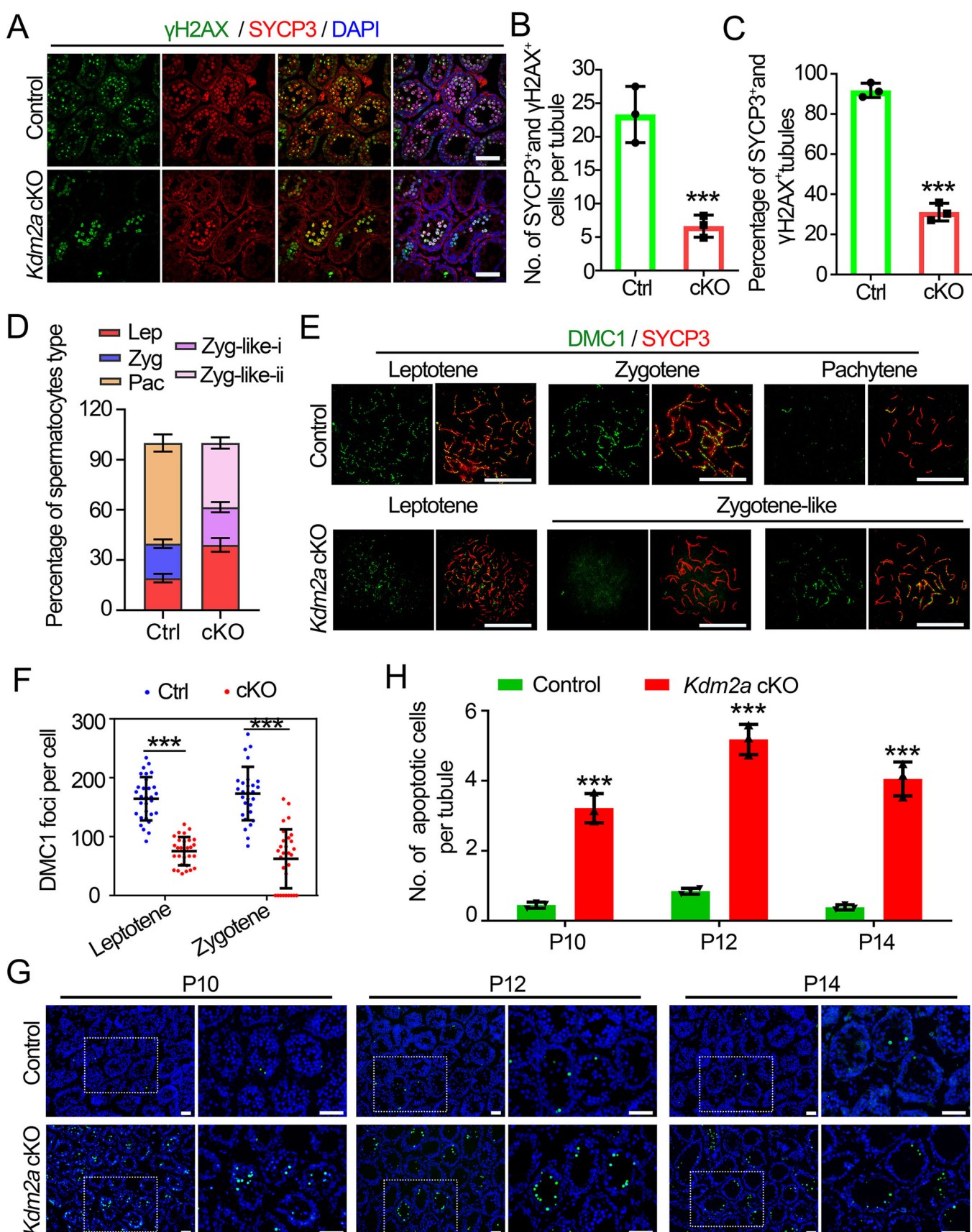

◄ **Figure EV4.** ***Kdm2a* deletion results in defective meiosis initiation and progression.**

(**A**) Co-immunostaining of SYCP3 with γH2AX in testis sections from control and *Kdm2a* cKO mice at P14. Nuclei were stained with DAPI. Scale bars = 50 μm. (**B, C**) The quantifications of SYCP3$^+$ and γH2AX$^+$ cells per tubule (**B**) and the percentage of SYCP3$^+$ and γH2AX$^+$ tubule (**C**) are shown. Data were presented as the mean ± SD. $n = 3$ biological replicates. *P* value was calculated using a two-tailed Student's *t*-test. ns, not significant. For (**B**), ***$P = 0.003$; For (**C**), ***$P < 0.0001$. (**D**) The percentage of spermatocytes at the leptotene (Lep), zygotene (Zyg), and pachytene (Pac) stages for (Fig. 3D). Data were presented as the mean ± SD. $n = 3$ biological replicates. (**E**) Spermatocyte spreads from control and *Kdm2a* cKO testes at P18 were co-stained SYCP3 and DMC1. Scale bars = 50 μm. (**F**) A scatter plot shows the number of DMC1 foci per cell on SYCP3 axes in leptotene and zygotene spermatocytes from control and *Kdm2a* cKO mice, respectively. Data are presented as the mean ± SD. A total of $n = 30$ control leptotema, $n = 29$ *Kdm2a* cKO leptotema, $n = 29$ control zygonema, and $n = 30$ *Kdm2a* cKO zygonema were counted from three biologically independent mice for each genotype. *P* value was calculated using a two-tailed Mann–Whitney *U*-test. left to right: ***$P < 0.0001$, ***$P < 0.0001$. (**G**) TUNEL staining of testis sections from control and *Kdm2a* cKO mice at P10, P12, and P14. Nuclei were stained with DAPI. Scale bars = 50 μm. (**H**) Quantification of apoptotic cells per tubule for (**G**). Data were presented as the mean ± SD. $n = 3$ biological replicates. *P* value was calculated using a two-tailed Student's *t*-test. left to right: ***$P = 0.0003$, ***$P < 0.0001$, ***$P = 0.0002$. Source data are available online for this figure.

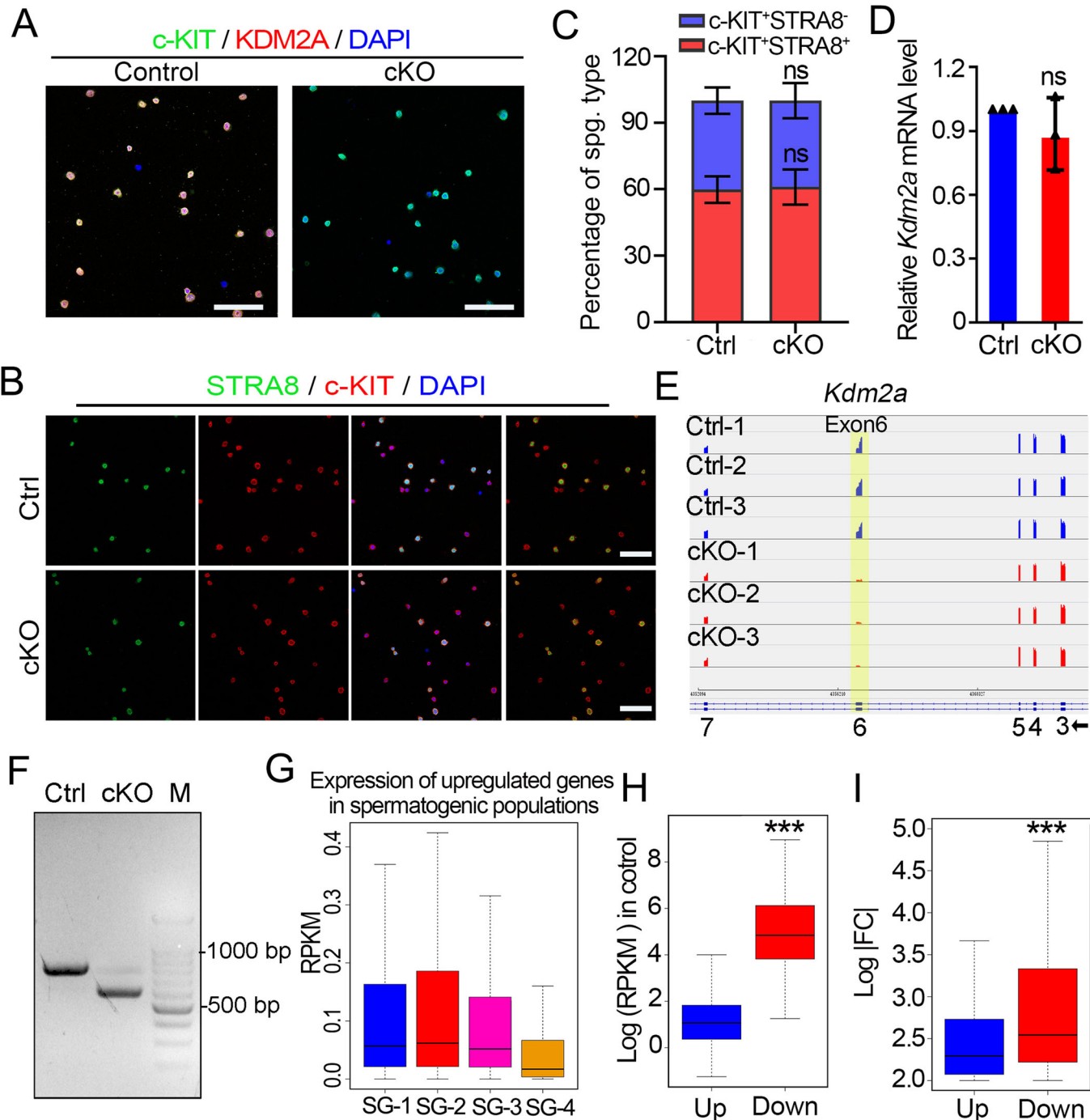

**Figure EV5.   Verification of KDM2A deletion in isolated c-KIT cells from *Kdm2a* cKO testes.**

(**A**) Co-immunofluorescence staining of anti-c-KIT with anti-KDM2A on isolated differentiated spermatogonal from control and KDM2A cKO mice. Nuclei were stained with DAPI. Scale bars = 50 μm. (**B**) Co-immunofluorescence staining of anti-STRA8 with anti-c-KIT on isolated differentiated spermatogonal from P10 control and *Kdm2a* cKO mice. Nuclei were stained with DAPI. Scale bars = 50 μm. (**C**) The percentage of c-KIT⁻ STRA8⁺ and c-KIT⁺STRA8⁺ spermatogonia from control and *Kdm2a* cKO mice. Data were presented as the mean ± SD. $n = 3$ biological replicates. *P* value was calculated using a two-tailed Student's *t*-test. ns, not significant. (**D**) RT-qPCR analyses of *Kdm2a* mRNA levels in isolated differentiated spermatogonal from P10 control and *Kdm2a* cKO mice. Data were presented as the mean ± SD. $n = 3$ biological replicates. *P* value was calculated using a two-tailed Student's *t*-test. ns, not significant. (**E**) Genome browser tracks depicting peaks of the region containing exon6 of Kdm2a based on RNA-seq. (**F**) RT-PCR analyses of *Kdm2a* mRNA in isolated c-KIT spermatogonia from control and *Kdm2a* cKO mice. (**G**) Expression levels (RPKM) of the upregulated 1934 genes in different types of spermatogonia (SG1-4) were reanalyzed with the previously published data of stage-specific bulk RNA-seq ($N = 3$ biologically independent samples). Whiskers indicate min and max. Bounds of box indicate 25th and 75th percentiles quantile with median. SPG 1, 2, 3, 4 correspond to SSCs (spermatogenic stem cells), undifferentiated spermatogonia, early differentiating spermatogonia and late differentiating spermatogonia, respectively. (**H**) Expression levels (RPKM) of the 1934 upregulated and 1922 downregulated genes in control (c-KIT+ cells of P10 control mice) are shown by box-whisker plot (whiskers indicate min and max. Bounds of box indicate 25th and 75th percentiles quantile with median). Data were presented as the mean ± SD. *P* value was calculated using an unpaired two-tailed Student's *t*-test. ***$P < 0.0001$. (**I**) Fold change of the 1934 upregulated and 1922 downregulated genes from RNA-seq are shown by box-whisker plot (whiskers indicate min and max. Bounds of box indicate 25th and 75th percentiles quantile with median). Data were presented as the mean ± SD. *P* value was calculated using an unpaired two-tailed Student's *t*-test. ***$P < 0.0001$. Source data are available online for this figure.

