## [Peer Review File · The EMBO Journal]

Histone demethylase KDM2A recruits HCFC1 and E2F1 to orchestrate male germ cell meiotic entry and progression

Shenglei Feng, Yiqian Gui, Shi Yin, Xinxin Xiong, Kuan Liu, Jinmei Li, Juan Dong, Xixiang Ma, Shunchang Zhou, Bingqian Zhang, Shiyu Yang, Fengli Wang, Xiaoli Wang, Xiaohua Jiang, and Shuiqiao Yuan

Corresponding authors: Shuiqiao Yuan (shuiqiaoyuan@hust.edu.cn) , Xiaohua Jiang (biojxh@ustc.edu.cn)

Review Timeline:

Submission Date:	16th Jan 24
Editorial Decision:	12th Feb 24
Revision Received:	2nd Jun 24
Editorial Decision:	14th Jul 24
Revision Received:	26th Jul 24
Accepted:	2nd Aug 24

Editor: Ieva Gailite

Transaction Report:

Dear Dr. Yuan,

Thank you for submitting your manuscript to The EMBO Journal. We have now received comments from three reviewers, which are included below for your information.

As you will see from the reports, the reviewers find the study interesting, while also indicating several concerns that would need addressing before they can recommend acceptance of the manuscript. In particular, reviewer #1 highlights that the proposed specific effect of KDMA2 on meiosis progression vs a more general differentiation defect would need further strengthening. The reviewers also raise instances of contradictory statements and results, as well as missing methodology description that would need to be clarified.

I find the raised points generally reasonable. Therefore, I would like to invite you to address the comments of the reviewers in a revised version of the manuscript. I should add that it is The EMBO Journal policy to allow only a single major round of revision and that it is therefore important to resolve the main concerns at this stage. Due to the more far-reaching issues raised by reviewer #1, I think that it would be useful to discuss the revision in more detail via email or phone/videoconferencing - please let me know which option you prefer.

We generally allow three months as standard revision time. As a matter of policy, competing manuscripts published during this period will not negatively impact on our assessment of the conceptual advance presented by your study. However, please contact me as soon as possible upon publication of any related work to discuss the appropriate course of action. Should you foresee a problem in meeting this deadline, please let us know in advance to discuss an extension.

When preparing your letter of response to the referees' comments, please bear in mind that this will form part of the Review Process File and will therefore be available online to the community. For more details on our Transparent Editorial Process, please visit our website: <https://www.embopress.org/page/journal/14602075/authorguide#transparentprocess>. Please also see the attached instructions for further guidelines on preparation of the revised manuscript.

Please feel free to contact me if have any further questions regarding the revision. Thank you for the opportunity to consider your work for publication, and I look forward to discussing your revision with you.

Yours sincerely,

Ieva Gailite

- a point-by-point response to the referees' comments, with a detailed description of the changes made (as a word file).
 - a word file of the manuscript text.
 - individual production quality figure files (one file per figure)
 - a complete author checklist, which you can download from our author guidelines (<https://www.embopress.org/page/journal/14602075/authorguide>).
 - Expanded View files (replacing Supplementary Information)
- Please see out instructions to authors
<https://www.embopress.org/page/journal/14602075/authorguide#expandedview>

We realize that it is difficult to revise to a specific deadline. In the interest of protecting the conceptual advance provided by the work, we recommend a revision within 3 months (12th May 2024). Please discuss the revision progress ahead of this time with the editor if you require more time to complete the revisions.

Referee #1:

In this paper, the authors studied a functional role of KDM2A in mitosis to meiosis transition. The authors showed that in the absence of KDM2A, meiosis progression was severely defective. Cytological and transcriptome data on c-Kit+ spermatogonial population have shown that there is a primary defect in cell differentiation. Furthermore, by their various omics data, the authors proposed that KDM2A directly regulates the expression of meiosis-related genes through the demethylation of histone at H3K36. Those analyses in mice corroborate in part the observed phenotype in the Kdm2a KO, and would provide an insight into the mechanisms of how KDM2A regulates H3K36me2/3 and gene expression network in spermatogenesis. Thus, their data is rich and the manuscript is potentially interesting to the field.

However, data presentation is yet to be sufficient without appropriate clarifications. Major concerns in this manuscript are raised from some gaps between the data and their assumption. Accordingly, some of the data are difficult to interpret, raising several concerns as described below. One of the major concerns is raised from their transcriptome analysis where their data may show population difference rather than bona fide DEG at the stage when primary defect appeared. Therefore, the later interpretations for the combined analyses of DEG by RNA-seq and KDM2A/H3K36me3 binding should be considered carefully, which needs further supporting data and rewriting the way of presentation. In particular, the direct effect of KDM2A on germ cell deficiency is still unclear. This reviewer does not doubt an importance of KDM2A in spermatogenic cell differentiation. However, the authors' conclusion that the meiosis is directly affected by KDM2A awaits further comprehensive analyses. Therefore, the current version of manuscript is overall preliminary to conclude the molecular function of KDM2A in meiosis, unless the authors could present additional supporting data and discuss clearly on the following concerns.

Fig1 I: please indicate which stage of spermatocyte? As the authors stated, since KDM2A level decreased during meiotic prophase progression, H3K36me1 and me3 levels may be different at different stages .

Line 182-185, 188-189 : The authors stated that "PLZF-STRA8+ preleptotene spermatocytes was significantly reduced in the cavity of seminiferous tubules relative to the littermate controls (Figure 2I-L), which indicated that spermatogonia differentiation was not affected, but meiosis initiation was impaired in Kdm2a cKO males". Since Kdm2A should be disrupted at the first time of STRA8 expression, it is uncertain exactly when the primary defect appeared. A more simple interpretation is that the spermatogonial differentiation was impaired or B-type spermatogonia was not maintained in Kdm2a cKO testis rather than that meiosis initiation was impaired. Please clarify the relevant sentences.

Lines 186-187: Here, the authors described a decrease in the number of c-KIT+ spermatogonia in KDM2A-deleted testes. This is reasonable as the spermatogonial population showed the strong expression of KDM2A in this population. However, the authors did not mention about this observation later on, even though they used cKIT+ cells for RNA-seq and ChIP-seq. Without the detailed analysis of KDM2A function in cKIT+ spermatogonial cell differentiation, it would be difficult to interpret the potential function of KDM2A in meiosis. Please clarify this issue.

Fig3B D : Quantification is not appropriate. Since some tubules have PLZF-/STRA8+/SYCP3+ while other tubules so not have

those cells at all, presentation of the average is not informative. Number of tubules that have PLZF-/STRA8+/SYCP3+ cells per total tubules should be shown rather than Number of PLZF-/STRA8+/SYCP3+ cells per tubules. The same way should be applied to PLZF-/STRA8+/rH2AX+ cells.

Line 206 : The authors stated that "the zygotene-like spermatocytes in Kdm2a cKO either lacked SYCP1 or exhibited abnormal accumulation in sister chromatids (Figure 3G)". Please indicate which is abnormal accumulation in sister chromatids, and explain how they are identified as sister chromatids.

Fig3F : Quantification should be shown.

Fig 4 : One concern in this manuscript could be raised from their transcriptome analysis where their data may represent population difference as the authors showed, rather than comparable DEG in WT vs cKO. Although they performed RNA-seq using c-Kit + enriched cells, it is uncertain whether transcriptome analysis was comparably done in WT vs cKO without population bias before primary defect appeared in the cKO. Therefore, the later interpretations for the combined analyses of DEG and KDM2A/H3K36me3 ChIP should be done carefully. The authors should clarify this issue and state technical limitation.

Line 215-217, Supplementary Figure 2K : Quantification should be shown.

Supplementary Figure 3D. : Please indicate the age of mice for the testis extracts. Is this done using whole testis? If so, protein levels apparently look different by means of different cellular population between the control and cKO testes.

Lines 242-247: The authors showed the mRNA and protein abundance of several known STRA8 target genes. REC8 expression is regulated independently of STRA8 as shown by Soh et al., PLOS genetics in 2015 or others. Although they showed the protein level of REC8 in supplementary Fig.3D, it is important to show the both Rec8 mRNA and protein abundance for proper understanding of the characteristics of KDM2A-deleted cells.

Line 323-328 : The authors stated that "633 genes that were up-regulated in Kdm2a cKO mice and bound by KDM2A overlapped with those genes with decreased H3K36me3 deposited in the Kdm2a cKO cells". This reviewer is confused. Are those 633 KDM2A-target genes showed decreased H3K36me3 level although those 633 KDM2A-target genes were defined as increased H3K36me3 in Kdm2a KO by ChIP-seq signal? Please clarify.

The same is applied to "516 genes that were down-regulated in Kdm2a cKO mice and bound by KDM2A overlapped with those genes with decreased H3K36me3 deposited in the Kdm2a cKO cells".

Fig5: Those authors stated that KDM2A-, H3K36me2-, H3K36me3-bound sites were identified by ChIP-seq. However, those experiments were done by CUT&RUN in the method. The authors should state accurately and more precisely what they indeed performed in this study.

Lines 387-417, Fig7C-J: Why do the authors only analyze the downregulated genes in Kdm2a-KO cells? They should also analyze the upregulated genes as well to see the effect of KDM2A, E2F and HCFC1 on expression.

Fig7A B: This reviewer wonders whether those experiments were indeed done by ChIP since no description of ChIP is found in the method.

Line 505-506 : " conditional knockout of KDM2A in male germ cells elevated meiotic genes and suppressed spermatogonial haploid genes at mitosis-to-meiosis transition stages" This sentence is opposite to their claim in the result session.

Minor concerns

Line 103-104 ; Please show a reference for KDM2A.

Line 123-126 ; "Supplementary Figure 2A, B" should be "Supplementary Figure 1A, B" . "Supplementary Figure 2C, D" should be "Supplementary Figure 1 C, D".

Lines 124: expression level is confusing here. As the authors used whole testes in Sup Fig.1C and D, it is better to re-written "the mRNA and protein expression level" to "the abundance of mRNA and protein in the tissue".

Fig.4 G and H: What do the colored terms mean? Please explain that in the legend.

Lines 352: The reference is focused on oogenesis but not spermatogenesis.

Referee #2:

In this manuscript by Feng, Gui and Yin et al., titled "KDM2A maintains H3K36me2/3 deposition and recruits HCFC1 and E2F1 to orchestrate male meiotic entry and progression", the authors describe the role of the histone demethylase, KDM2A, during spermatogenesis. The authors demonstrate that KDM2A is required for proper meiotic gene expression. They show that KDM2A binds the promoter regions of a broad spectrum of genes in germ cells, including meiotic genes. Conditional loss of KDM2A in germ cells disrupts the balance of H3K36me2 and H3K36me3. KDM2A loss also leads to aberrant gene expression of several classes of genes, including meiotic prophase genes, culminating in cell death just prior to and during early meiotic prophase. Authors identify the transcription factor E2F1 and its co-factor HCF1 as protein-interacting partners of KDM2A and propose that these three factors collaborate to orchestrate gene expression regulation during spermatogenesis.

This manuscript provides new mechanistic insight into the regulation of meiotic gene expression and the research is suitable for publication in EMBO Journal. While I have no major comments on the research, the text needs several corrections (see minor comments below).

Minor comments

1. There are typos/errors throughout the manuscript. For example, PRDM9 is misspelled on page 4, line 87. On page 5, line 123-124, the authors refer to supplementary figure 2A-B, instead of 1A-B. The sentence on page 4, line 110 does not make sense. Figure legend 1E-F states that P10 mice were used, whereas the text states that P14 mice were used. In figure 3E, the bars are colored incorrectly. Also, figure 5N, O have been mis-referenced as 4N, O in the text on page 12. There are other errors that I have not listed out. I suggest thoroughly proofreading.
2. It would be useful for a reader to know the cell stage at which Stra8-Cre is expected to act. In particular, the authors should define the stage at which KDM2A is lost in their system. What was the expression level of Kdm2a in cKIT+ cKO RNA sequencing data?
3. Were the data presented in supplementary figure 3B, C, D obtained from whole testes of juvenile mice? This information should be included in the main text as cell population differences may account for some of these differences (i.e., loss of meiotic cells in the mutant may explain why there are less meiotic RNAs and proteins).
4. In figure 1C, what does SPG 1, 2, 3, 4 refer to? Please explain in figure legend.
5. The authors use the term "deposition" to describe the effects of KDM2A disruption on me2/3. I suggest using "levels": KDM2A ablation leads to abnormal levels of H3K36me2/3 in germ cells as this is more consistent with the function that they describe.

Referee #3:

In this manuscript, the authors demonstrated the role of histone demethylase KDM2A in male germ cell development. They present the spatio-temporal expression pattern of KDM2A during the transition from mitosis to meiosis and meiotic progression during spermatogenesis. They utilized germ cell-specific Kdm2a ablation to investigate the role of KDM2A in spermatogenesis. They report changes in the gene mitosis/meiosis expression program along with changes in H3K36 methylation levels, resulting in an aberrant mitosis-to-meiosis transition. Additionally, they observed a co-dependence between KDM2A and the transcription factors E2F1 and HCFC1. These results suggest a mechanism of KDM2A-dependent gene regulation in differentiating spermatogonia.

The study is well-designed, with detailed analyses supporting a potentially novel function of KDM2A in testis along with correlative evidence implicating its role in recruiting other transcription factors. Most of the data presented were robust and supported their conclusions. The following are suggestions for consideration.

The comparison of H3K36me3 levels at KDM2A-bound genes in the results section is confusing and seemingly contradictory. The levels are described as 'decreased' (lines 325 to 327), while Figures 5 N-O show increased levels. Please revise or explain. Supplementary Figure 4 shows 3102 KDM2a-targeted genes with an increase in H3K36me3. In contrast, 3107 KDM2a-targeted genes show a reduction in H3K36me2 without an increase in H3K36me3. Please provide an explanation for the lack of increase in trimethylation.

Although Figure 6C shows a positive co-IP band for E2F1, supplementary Figure 6A also shows bands of similar intensity for E2F1 in both WT and Kdm2a-cKO c-KIT positive cells. One could conclude that the KDM2A antibody used in the IP experiments non-specifically binds to E2F1 and, therefore, brings down a similar amount of E2F1 in KDM2A cKO cells. This comment should be addressed.

Single-cell RNAseq data analysis should be included in the methods section.

The figure references in the results section are incorrect in several places, eg. Lines 123, 126, 326, 328. Legends in figure panels (eg. Fig 3E) should also be revised.

POINT-BY-POINT RESPONSE TO REVIEWERS

GENERAL COMMENTS FOR ALL REVIEWERS

We greatly thank the Editor and all Reviewers for their careful consideration of our manuscript (EMBOJ-2024-116688) and for their valuable comments, which enabled us to significantly improve and highlight the importance of our manuscript. All reviewers found our work to be interesting to the field but raised several concerns regarding our manuscript. We have performed many additional experiments and addressed all the comments raised by all three reviewers. We have also revised the confusing descriptions and corrected all typos/errors. We sincerely hope that all Reviewers will be satisfied with our revision and find it suitable for publication.

Our specific comments to each reviewer follow. Please note: Reviewer comments are in italics. Our responses are in blue font. The comments from the Reviewers have not been edited. We thank you again for your feedback and consideration.

REVIEWER COMMENTS

Referee #1:

In this paper, the authors studied a functional role of KDM2A in mitosis to meiosis transition. The authors showed that in the absence of KDM2A, meiosis progression was severely defective. Cytological and transcriptome data on c-KIT+ spermatogonial population have shown that there is a primary defect in cell differentiation. Furthermore, by their various omics data, the authors proposed that KDM2A directly regulates the expression of meiosis-related genes through the demethylation of histone at H3K36. Those analyses in mice corroborate in part the observed phenotype in the Kdm2a KO, and would provide an insight into the mechanisms of how KDM2A regulates H3K36 me2/3 and gene expression network in spermatogenesis. Thus, their data is rich and the manuscript is potentially interesting to the field.

Response: We appreciate that the reviewer finds our work interesting and useful for the field. We also thank him/her for the critical feedback to further improve our manuscript.

However, data presentation is yet to be sufficient without appropriate clarifications. Major concerns in this manuscript are raised from some gaps between the data and their assumption. Accordingly, some of the data are difficult to interpret, raising several concerns as described below. One of the major concerns is raised from their transcriptome analysis where their data may show population difference rather than bona fide DEG at the stage when primary defect appeared. Therefore, the later interpretations for the combined analyses of DEG by RNA-seq and KDM2A/H3K36me3 binding should be considered carefully, which needs further supporting data and rewriting the way of presentation. In particular, the direct effect of KDM2A on germ cell deficiency is still unclear. This reviewer does not doubt an importance of KDM2A in spermatogenic cell differentiation. However, the authors' conclusion that the meiosis is directly affected by KDM2A awaits further comprehensive analyses. Therefore, the current version of manuscript is overall preliminary to conclude the molecular function of KDM2A in meiosis, unless

the authors could present additional supporting data and discuss clearly on the following concerns.

Response: Thank you for your constructive comments and suggestions, which enabled us to improve the quality of the manuscript. In this revision, we have performed many extra experiments and presented additional supporting data to make our conclusions more convincing. Specifically, we tested the purity of c-KIT⁺ cells isolated from juvenile (P10) mouse testes, and the new results showed that the purity of c-KIT⁺ cells was over 90% in both the control and cKO groups (new **Fig. EV5A**), ruling out the possibility that the DEGs observed in the cKO group were caused by cell population differences. We also performed several additional sets of biological replicates on the phenotype analyses of *Kdm2a* cKO mice and confirmed that the differentiation of spermatogonia during the first round of spermatogenesis was not affected by KDM2A ablation and the primary defect we observed was the significant decrease in the number of preleptotene spermatocytes in the *Kdm2a* cKO mice (new **Fig. 3A-D**), suggesting that meiotic initiation was severely affected by *Kdm2a* deletion. In addition, we did not find any significant differences in the number of early (A1 to A4) and late (intermediate to B-type) differentiating spermatogonia between the control and cKO groups (**Fig. EV2C-E**), further demonstrating that KDM2A functions in meiosis and not in spermatogonial differentiation during the first wave of spermatogenesis. As all samples used for RNA-seq in this study are the purified c-KIT⁺ spermatogonia from juvenile (P10) mouse testes, the possibility of the effects of spermatogonia population differences is further excluded. We have addressed all your concerns by providing additional supporting data and reorganizing the manuscript. We thank you again for your comments on improving the quality of our manuscript and hope that you are satisfied with our revision.

Fig1 I: please indicate which stage of spermatocyte? As the authors stated, since KDM2A level decreased during meiotic prophase progression, H3K36me1 and me3 levels may be different at different stages.

Response: Thank you for your comments. To further determine the exact expression of H3K36me1/2/3 at which stage of the spermatocyte, we performed a chromosome spreading assay on different stages of spermatocytes. The new results showed that H3K36me1 and H3K36me2 were highly expressed in leptotene and zygotene spermatocytes, but showed low expression in pachytene and diplotene spermatocytes. On the contrary, the expression level of H3K36me3 was significantly higher in pachytene and diplotene spermatocytes compared to the early meiotic stages (see **Fig.EV1I**). Therefore, the expression pattern of H3K36me1/2 was consistent with KDM2A but the H3K36me3 was different with KDM2A expression. We have included the new results in the revision (see lines 160-166 with yellow highlighted).

Line 182-185, 188-189: The authors stated that "PLZF-STRA8+ preleptotene spermatocytes was significantly reduced in the cavity of seminiferous tubules relative to the littermate controls (Figure 2I-L), which indicated that spermatogonia differentiation was not affected, but meiosis initiation was impaired in Kdm2a cKO males". Since Kdm2A should be disrupted at the first time of STRA8 expression, it is uncertain exactly when the primary defect appeared. A more simple interpretation is that the spermatogonial differentiation was impaired or B-type spermatogonia was not maintained in Kdm2a cKO testis rather than that meiosis initiation was impaired. Please clarify the relevant sentences.

Response: Thank you for asking this critical question. Based on our IF results, KDM2A was indeed knocked out in P5 testes, which is the time point prior to STRA8 expression (**Fig. EV2B**). However, as answered above, our new data showed that the differentiation of spermatogonia was not affected in *Kdm2a* cKO testes at P10, but most germ cells were unable to enter meiosis during the first wave of spermatogenesis (**Fig. 3A-D**). In addition, by co-staining with STRA8 and c-KIT in P10 testes, we found no significant difference in the number of STRA8⁺ c-KIT⁺ cells (A1 to A4) and the number of STRA8⁻ c-KIT⁺ cells (intermediate to B type) between the control and cKO groups (**Fig. EV2C-E**). Therefore, these additional data further demonstrated that *Kdm2a* deletion does not affect spermatogonial differentiation but disrupts meiotic initiation and progression. We have added the new data and also revised the relevant sentences in the revision (see lines 186-204 with yellow highlighted).

Lines 186-187: Here, the authors described a decrease in the number of c-KIT⁺ spermatogonia in KDM2A-deleted testes. This is reasonable as the spermatogonial population showed the strong expression of KDM2A in this population. However, the authors did not mention about this observation later on, even though they used cKIT⁺ cells for RNA-seq and ChIP-seq. Without the detailed analysis of KDM2A function in cKIT⁺ spermatogonial cell differentiation, it would be difficult to interpret the potential function of KDM2A in meiosis. Please clarify this issue.

Response: Thank you for your comments. In fact, we observed that the number of c-KIT⁺ spermatogonia decreased only in adult *Kdm2a* cKO testes, not in juvenile testes (P10), suggesting that KDM2A could affect spermatogonial differentiation in steady-state spermatogenesis, not in first-wave spermatogenesis (**Fig. EV2D-I**). To confirm this, we generated new *Kdm2a*^{flox/flox}; *Ddx4*-Cre^{ERT2} male mice for tamoxifen-induced KDM2A deletion (iKO) in germ cells to investigate whether the spermatogonial differentiation process is affected by *Kdm2a* ablation in adulthood. The new data showed that the iKO mice exhibited a decrease in the number of c-KIT⁺ spermatogonia and a comparable number of PLZF⁺ spermatogonia, which phenocopied the adult *Kdm2a* cKO mice (**Fig. EV3D-I**), further demonstrating that KDM2A is essential for spermatogonial differentiation during steady-state adult spermatogenesis. As all the samples used for sequencing analyses (RNA-seq, ChIP-seq, CUT&RUN-seq, etc.) in this study are the purified c-KIT⁺ spermatogonia from juvenile (P10) mouse testes, the possibility of the effects of spermatogonia population differences can be excluded. We have also included these new data and revised the statement in the revision (see lines 204-214 with yellow highlighted).

Fig3B D: Quantification is not appropriate. Since some tubules have PLZF-/STRA8+/SYCP3+ while other tubules so not have those cells at all, presentation of the average is not informative. Number of tubules that have PLZF-/STRA8+/SYCP3+ cells per total tubules should be shown rather than Number of PLZF-/STRA8+/SYCP3+ cells per tubules. The same way should be applied to PLZF-/STRA8+/rH2AX+ cells.

Response: We appreciate and agree with your great suggestion. As suggested, we have quantified the number of tubules that have PLZF-/STRA8+/SYCP3+ or PLZF-/STRA8+/γH2AX+ cells per total tubules and shown the percentage of tubules with PLZF-/STRA8+/SYCP3+ cells (**Fig.3B**) or PLZF-/STRA8+/γH2AX+ cells (**Fig.3D**) in the revised **Fig.3**. We have also described the quantification method in the figure legend of Fig.3B and D.

Line 206: The authors stated that "the zygotene-like spermatocytes in *Kdm2a* cKO either lacked SYCP1 or exhibited abnormal accumulation in sister chromatids (Figure 3G)". Please indicate which is abnormal accumulation in sister chromatids, and explain how they are identified as sister chromatids.

Response: Thank you for your comments. As shown in **Fig.3F-G**, we found two types of zygotene-like spermatocytes, represented by type-i and type-ii, respectively. By quantifying the number of chromosomes, it was found that there are a total of 40 chromosomes in type-i zygotene-like spermatocytes (see figure below), indicating that these are sister chromatids without synapsis. Normally, SYCP1 is only located on synapsed homologous chromosomes; however, in *Kdm2a* cKO mice, it accumulated abnormally on unsynapsed sister chromatids, as indicated by the arrowheads in **Fig.3G**.

Fig3F: Quantification should be shown.

Response: Done as suggested (see new **Fig. EV4D**).

Fig 4: One concern in this manuscript could be raised from their transcriptome analysis where their data may represent population difference as the authors showed, rather than comparable DEG in WT vs cKO. Although they performed RNA-seq using c-KIT⁺ enriched cells, it is uncertain whether transcriptome analysis was comparably done in WT vs cKO without population bias before primary defect appeared in the cKO. Therefore, the later interpretations for the combined analyses of DEG and KDM2A/H3K36me3 ChIP should be done carefully. The authors should clarify this issue and state technical limitation.

Response: Thank you for your comments. As responded above, all RNA-seq and CUT/RUN-seq were performed using purified c-KIT⁺ cells from juvenile (P10) mouse testes. In this revision, we have tested the purity of the isolated cells and found that the purity of c-KIT⁺ cells can reach >90% (**Fig. EV5A**). In addition, we co-stained with STRA8 and c-KIT, and found that the proportion of early differentiating spermatogonia (A1 to A4 indicated by STRA8⁺c-KIT⁺) and late differentiating spermatogonia (A1 to A4 indicated by STRA8⁻c-KIT⁺) was comparable in the control and cKO groups (**Fig. EV5B-C**). These new results further confirmed that the high purity c-KIT⁺ cells isolated from WT and cKO juvenile mouse testes at P10 were comparable between the WT control and cKO groups without population bias before the primary meiotic defect appeared in the cKO mice, further supporting our subsequent RNA-seq/CUT&RUN-seq analyses of purified

c-KIT⁺ cells as bona fide DEGs not derived from cell population bias. We have also added the new data in the revision and mentioned it in the text for clarification (see lines 249-254 with yellow highlighted).

Line 215-217, Supplementary Figure 2K: Quantification should be shown.

Response: Thank you for your careful review. We have made quantification for the apoptotic cells (see new **Fig.EV4H**).

Supplementary Figure 3D.: Please indicate the age of mice for the testis extracts. Is this done using whole testis? If so, protein levels apparently look different by means of different cellular population between the control and cKO testes.

Response: Thank you for your question. In fact, we performed the WB assay using P10 purified germ cell extracts rather than whole testis extracts to avoid differences in protein expression caused by the different cell population between control and cKO testes. We have shown the results in the new **Appendix Fig. 1C** (previously Supplementary Figure 3D).

Lines 242-247: The authors showed the mRNA and protein abundance of several known STRA8 target genes. REC8 expression is regulated independently of STRA8 as shown by Soh et al., PLOS genetics in 2015 or others. Although they showed the protein level of REC8 in supplementary Fig.3D, it is important to show the both Rec8 mRNA and protein abundance for proper understanding of the characteristics of KDM2A-deleted cells.

Response: We appreciate and agree with your comments. In this revision, we have determined the mRNA level of *Rec8* by qPCR and found that *Rec8* mRNA was downregulated in KDM2A-deleted cells (**Appendix Fig. 1B**), which is consistent with our RNA-seq data. Therefore, our results, combined with previously published data (Soh et al, 2015), suggest that *Rec8* expression in spermatogenic cells may be regulated by *Kdm2a* independently of *Stra8*.

Line 323-328: The authors stated that "633 genes that were up-regulated in Kdm2a cKO mice and bound by KDM2A overlapped with those genes with decreased H3K36me3 deposited in the Kdm2a cKO cells". This reviewer is confused. Are those 633 KDM2A-target genes showed decreased H3K36me3 level although those 633 KDM2A-target genes were defined as increased H3K36me3 in Kdm2a KO by ChIP-seq signal? Please clarify.

The same is applied to "516 genes that were down-regulated in Kdm2a cKO mice and bound by KDM2A overlapped with those genes with decreased H3K36me3 deposited in the Kdm2a cKO cells".

Response: Thank you for your comments. In these sentences, we wanted to express that those 633 KDM2A-target upregulated genes showed increased H3K36me3 occupancy in *Kdm2a* cKO cells. We have revised the description accordingly in the revision (see lines 364-368 with yellow highlighted).

Fig5: Those authors stated that KDM2A-, H3K36me2-, H3K36me3-bound sites were identified by ChIP-seq. However, those experiments were done by CUT&RUN in the method. The authors should state accurately and more precisely what they indeed performed in this study.

Response: Thank you for your careful review. We did indeed perform CUT&RUN, not ChIP, to

identify KDM2A, H3K36me2, H3K36me3 bound sites in our study. We have corrected all descriptions in the revision and apologize for our oversight.

Lines 387-417, Fig7C-J: Why do the authors only analyze the downregulated genes in Kdm2a-KO cells? They should also analyze the upregulated genes as well to see the effect of KDM2A, E2F and HCFC1 on expression.

Response: Thank you very much for your question and suggestion. As suggested, in this revision we selected several upregulated genes bound by KDM2A, E2F1 and HCFC1 for CUT&RUN qPCR analyses and found that most target genes showed decreased occupancy of both E2F1 and HCFC1 (**Appendix Fig. 4B, C**). We then selected four genes (*Sox3*, *Trim71*, *Sall4* and *Eomes*) to perform a dual luciferase reporter assay. The result showed that the expression of these four genes can be suppressed by KDM2A and E2F1, but not by HCFC1 (**Appendix Fig.4J-Q**), suggesting that KDM2A and E2F1 may also be involved in regulating the transcription of these upregulated genes. We have included the new results in the revision (see lines 454-460 with yellow highlighted).

Fig7A B: This reviewer wonders whether those experiments were indeed done by ChIP since no description of ChIP is found in the method.

Response: As responded to above, we performed CUT&RUN rather than ChIP in this study and have corrected all descriptions in the revision. Thank you very much again.

Line 505-506: " conditional knockout of KDM2A in male germ cells elevated meiotic genes and suppressed spermatogonial haploid genes at mitosis-to-meiosis transition stages" This sentence is opposite to their claim in the result session.

Response: Thank you for your careful review. We have corrected this sentence in the revision (see lines 563-564 with yellow highlighted).

Minor concerns

Line 103-104; Please show a reference for KDM2A.

Response: We have added a review article (Liu *et al*, 2021) as a reference (lines 106-107).

Line 123-126; "Supplementary Figure 2A, B" should be "Supplementary Figure 1A, B" . "Supplementary Figure 2C, D" should be "Supplementary Figure 1 C, D".

Response: We have corrected it. Now it is named as **Fig.EV1A, B** and **Fig.EV1C, D** in the revision to meet with the Journal format requirements. Thank you for your careful review.

Lines 124: expression level is confusing here. As the authors used whole testes in Sup Fig.1C and D,it is better to re-written "the mRNA and protein expression level" to "the abundance of mRNA and protein in the tissue".

Response: Done as suggested. Thank you!

Fig.4 G and H: What do the colored terms mean? Please explain that in the legend.

Response: These colored terms are key biological processes related to the development of spermatogenic cells. We have explained this in the legend to **Fig.4 G-H**.

Lines 352: The reference is focused on oogenesis but not spermatogenesis.

Response: Thank you for pointing out the incorrect reference. We have removed the previous reference to oogenesis and added two references (Jorgez *et al*, 2021; Rotgers *et al*, 2015) to the function of E2F1 in spermatogenesis.

Referee #2:

In this manuscript by Feng. Gui and Yin et al., titled "KDM2A maintains H3K36me2/3 deposition and recruits HCFC1 and E2F1 to orchestrate male meiotic entry and progression", the authors describe the role of the histone demethylase, KDM2A, during spermatogenesis. The authors demonstrate that KDM2A is required for proper meiotic gene expression. They show that KDM2A binds the promoter regions of a broad spectrum of genes in germ cells, including meiotic genes. Conditional loss of KDM2A in germ cells disrupts the balance of H3K36me2 and H3K36me3. KDM2A loss also leads to aberrant gene expression of several classes of genes, including meiotic prophase genes, culminating in cell death just prior to and during early meiotic prophase. Authors identify the transcription factor E2F1 and its co-factor HCF1 as protein-interacting partners of KDM2A and propose that these three factors collaborate to orchestrate gene expression regulation during spermatogenesis.

This manuscript provides new mechanistic insight into the regulation of meiotic gene expression and the research is suitable for publication in EMBO Journal. While I have no major comments on the research, the text needs several corrections (see minor comments below).

Response: Thank you so much for your appreciation of the significance of our work. We also appreciate your constructive suggestions, which helped us to improve the quality of the manuscript considerably.

Minor comments

1. There are typos/errors throughout the manuscript. For example, PRDM9 is misspelled on page 4, line 87. On page 5, line 123-124, the authors refer to supplementary figure 2A-B, instead of 1A-B. The sentence on page 4, line 110 does not make sense. Figure legend 1E-F states that P10 mice were used, whereas the text states that P14 mice were used. In figure 3E, the bars are colored incorrectly. Also, figure 5N, O have been mis-referenced as 4N, O in the text on page 12. There are other errors that I have not listed out. I suggest thoroughly proofreading.

Response: Thank you for pointing out these errors. We have corrected all the typos/errors throughout the manuscript according to your suggestions.

2. It would be useful for a reader to know the cell stage at which Stra8-Cre is expected to act. In particular, the authors should define the stage at which KDM2A is lost in their system. What was the expression level of Kdm2a in cKIT+ cKO RNA sequencing data?

Response: Thank you for your comments. Based on previous reports (Lin *et al*, 2017; Sadate-Ngatchou *et al*, 2008), it was expected that Stra8-Cre would exert its recombinase activity in type A1 spermatogonia as early as P3 testes. In our study, we found that Kdm2a was successfully knocked out in the germ cells of P5 mouse testes. (Fig. EV2B). We have mentioned the time point of Stra8-Cre activity in the revision (see lines 172-173).

Interestingly, *Kdm2a* did not appear in the DEGs and showed unchanged mRNA levels in the *Kdm2a* cKO cells by qPCR assay (**Fig. EV5D**); however, it showed exon-6 skipping based on the RNA-seq data viewed in the Integrative Genomic Viewer browser (**Fig. EV5E**). By RT-PCR analysis, we found that the *Kdm2a* transcript with the exon-6 deletion was still present in *Kdm2a* cKO germ cells (**Fig. EV5F**). These data suggest that Cre-mediated deletion of exon-6 of *Kdm2a* did not lead to degradation of this truncated transcript, despite the premature appearance of termination codons caused by the frame-shift mutation. In fact, we did not detect smaller KDM2A protein bands in *Kdm2a* cKO germ cells, suggesting that the non-functional protein translated from this truncated transcript may be degraded or unrecognizable by the KDM2A antibody. We have mentioned this in the revision for the readers' better understanding (lines 256-263 with yellow highlighted).

3. *Were the data presented in supplementary figure 3B, C, D obtained from whole testes of juvenile mice? This information should be included in the main text as cell population differences may account for some of these differences (i.e., loss of meiotic cells in the mutant may explain why there are less meiotic RNAs and proteins).*

Response: Thank you for your questions. In our study, we used purified c-KIT⁺ cells from juvenile mice (P10) for all sequencing/qPCR/WB analyses, which could effectively avoid confounding by differences in cell population. We have now included this information in the main text (lines 285-286).

4. *In figure 1C, what does SPG 1, 2, 3, 4 refer to? Please explain in figure legend.*

Response: In the revised manuscript, we have provided an explanation in the figure legend (**Fig.1C and EV5G**). SPG 1, 2, 3, 4 correspond to SSCs (spermatogenic stem cells), undifferentiated spermatogonia, early differentiating spermatogonia and late differentiating spermatogonia, respectively according to previous report (Wang *et al*, 2019).

5. *The authors use the term "deposition" to describe the effects of KDM2A disruption on me2/3. I suggest using "levels": KDM2A ablation leads to abnormal levels of H3K36me2/3 in germ cells as this is more consistent with the function that they describe.*

Response: Done as suggested. Thank you very much.

Referee #3:

*In this manuscript, the authors demonstrated the role of histone demethylase KDM2A in male germ cell development. They present the spatio-temporal expression pattern of KDM2A during the transition from mitosis to meiosis and meiotic progression during spermatogenesis. They utilized germ cell-specific *Kdm2a* ablation to investigate the role of KDM2A in spermatogenesis. They report changes in the gene mitosis/meiosis expression program along with changes in H3K36 methylation levels, resulting in an aberrant mitosis-to-meiosis transition. Additionally, they observed a co-dependence between KDM2A and the transcription factors E2F1 and HCFC1. These results suggest a mechanism of KDM2A-dependent gene regulation in differentiating spermatogonia.*

The study is well-designed, with detailed analyses supporting a potentially novel function of

KDM2A in testis along with correlative evidence implicating its role in recruiting other transcription factors. Most of the data presented were robust and supported their conclusions. The following are suggestions for consideration.

Response: Thank you for your appreciation of the novel function of KDM2A in germ cell development and your valuable suggestions, which enabled us to improve the quality of the manuscript.

The comparison of H3K36me3 levels at KDM2A-bound genes in the results section is confusing and seemingly contradictory. The levels are described as 'decreased' (lines 325 to 327), while Figures 5 N-O show increased levels. Please revise or explain.

Response: We have corrected it to 'increased', consistent with **Fig. 5N-O**. We thank you for your careful review and apologize for this error description.

Supplementary Figure 4 shows 3102 KDM2a-targeted genes with an increase in H3K36me3. In contrast, 3107 KDM2a-targeted genes show a reduction in H3K36me2 without an increase in H3K36me3. Please provide an explanation for the lack of increase in trimethylation.

Response: Thank you for your suggestions. We also noticed that almost half of the KDM2A-targeted genes with a reduction in H3K36me2 were not accompanied by an increase in H3K36me3 (**Appendix Fig. S3D**). There are two reasons that could explain this phenomenon:

1) Although our data showed that *Kdm2a* deletion caused an increase in H3K36me2 and a decrease in H3K36me3 in germ cells, we cannot exclude the possibility that KDM2A may function as a demethylase targeting H3K36me2, just like its traditional role in somatic cells, which can bind some genes and catalyze H3K36me2 deposited in these genes to H3K36me1.

2) In addition to *Kdm2a*, other H3K36 methylases and demethylases may also play a role in germ cell development. It is worth noting that the other two H3K36me3 demethylases (Sharda & Humphrey, 2022), *Kdm4a* and *Kdm4c*, were found to be up- and downregulated, respectively, in our *Kdm2a* cKO germ cells (**Table EV1**), suggesting that they may compete or cooperate with each other to regulate the balance of H3K36me2 and H3K36me3 distribution on KDM2A-targeted genes. However, we cannot rule out the involvement of other factors in this process, which will be worthy of further investigation in the future.

As suggested, we have included the above explanation in the Discussion section of the revision (see lines 542-555 with yellow highlighted).

Although Figure 6C shows a positive co-IP band for E2F1, supplementary Figure 6A also shows bands of similar intensity for E2F1 in both WT and Kdm2a-cKO c-KIT positive cells. One could conclude that the KDM2A antibody used in the IP experiments non-specifically binds to E2F1 and, therefore, brings down a similar amount of E2F1 in KDM2A cKO cells. This comment should be addressed.

Response: We apologize for mislabeling the name of the IP antibody in **Appendix Fig. 4A** (previous supplementary Figure 6A). In fact, we used the E2F1 antibody for IP, but not the KDM2A antibody. We have corrected this in **Appendix Fig. 4A**. This result suggests that the interaction of E2F1 with HCFC1 is affected by the deletion of KDM2A.

Single-cell RNAseq data analysis should be included in the methods section.

Response: We have included the method for analysis of single-cell RNA-seq data in the Methods section (see lines 786-795).

The figure references in the results section are incorrect in several places, eg. Lines 123, 126, 326, 328. Legends in figure panels (eg. Fig 3E) should also be revised.

Response: We have carefully checked the manuscript and have corrected all the typos/errors throughout the manuscript. Thank you for your careful review.

REFERENCES:

Jorgez CJ, Seth A, Wilken N, Bournat JC, Chen CH, Lamb DJ (2021) E2F1 regulates testicular descent and controls spermatogenesis by influencing WNT4 signaling. *Development* 148

Lin Z, Hsu PJ, Xing X, Fang J, Lu Z, Zou Q, Zhang KJ, Zhang X, Zhou Y, Zhang T *et al* (2017) Mettl3-/Mettl14-mediated mRNA N(6)-methyladenosine modulates murine spermatogenesis. *Cell Res* 27: 1216-1230

Liu L, Liu J, Lin Q (2021) Histone demethylase KDM2A: Biological functions and clinical values (Review). *Exp Ther Med* 22: 723

Rotgers E, Nurmio M, Pietila E, Cisneros-Montalvo S, Toppari J (2015) E2F1 controls germ cell apoptosis during the first wave of spermatogenesis. *Andrology-Us* 3: 1000-1014

Sadate-Ngatchou PI, Payne CJ, Dearth AT, Braun RE (2008) Cre recombinase activity specific to postnatal, premeiotic male germ cells in transgenic mice. *Genesis* 46: 738-742

Sharda A, Humphrey TC (2022) The role of histone H3K36me3 writers, readers and erasers in maintaining genome stability. *DNA Repair (Amst)* 119: 103407

Soh YQ, Junker JP, Gill ME, Mueller JL, van Oudenaarden A, Page DC (2015) A Gene Regulatory Program for Meiotic Prophase in the Fetal Ovary. *PLoS Genet* 11: e1005531

Wang ZP, Xu XJ, Li JL, Palmer C, Maric D, Dean J (2019) Sertoli cell-only phenotype and scRNA-seq define PRAMEF12 as a factor essential for spermatogenesis in mice. *Nature Communications* 10

Dear Dr. Yuan,

Thank you for submitting a revised version of your manuscript. I sincerely apologise for the protracted assessment process due to conference travel and the high number of submissions we receive at the moment.

Your study has now been seen by two of the original referees. Reviewer #3 finds that the previous concerns have been addressed satisfactorily, while reviewer #1 finds that softening of the conclusions regarding a direct role of KDM2A would be needed in the final version. Based on this referee input, I would like to invite you to submit a final revision in which the statements are toned down as requested by reviewer #1. Additionally, please address the editorial points below:

1. Please check that the funding information is correct and identical both in the manuscript and our online system. Currently, National Natural Science Foundation of China (82371625 to S.Y., 82101738 to J.D., and 82071709 to X.J.), the Joint Fund for Medical Artificial Intelligence (MAI2022Q010 to X.J.), the Key Research and Development Project of Anhui Province (2022e07020014 to X.J.), and the Basic Research Support Program of Huazhong University of Science and Technology (2023BR031 to S.Y.) are missing in our system. In the manuscript file, please include the funding information in "Acknowledgments" section.
2. CRediT has replaced the traditional author contributions section because it offers a systematic, machine-readable author contributions format that allows for more effective research assessment. Please remove the Authors Contributions from the manuscript and use the free text boxes beneath each contributing author's name in our online submission system to add specific details on the author's contribution. More information is available in our guide to authors.
3. Please rename Table EV1-EV5 into Dataset EV1-EV5 and upload their legends as a separate tab in each Excel file.
4. In the Data Availability section, please add a resolvable link for PRJNA1066023 dataset. More information about the format of this section can be found here: <https://www.embopress.org/page/journal/14602075/authorguide#dataavailability>.
5. During our routine text plagiarism check, we noted that numerous sentences in the manuscript shows high similarity to these from other publications - please see the attached screenshots. Please rephrase the text accordingly.
6. In our standard image integrity check, we noted that in Figure 1D, the same image has been re-used in the Zygotene St.XII and Diplotene St.XII panels. Please clarify. If this is intentional, please note the image reuse in the figure legend.
7. The source data do not appear to fit to the data in Fig. 2K (Kdm2a cKO) - please check.
8. In the figure panel 5H, H3K36me1 blot, the background signal does not match to that in the source data and appears to have been removed. Please note that selective removal of the signal is not allowed. Please replace the image accordingly with one where the background signal matches the original data.
9. Please complete the provided Source Data checklist; source data files need to be reorganised to one file/folder per figure. For main figures, please provide one ZIP folder per main figure. For EV and/or appendix figures, please zip together all source data.
10. Our data editors have flagged the following issues in figure legends that need correcting:
 - Please provide the exact p values in the legends of figures 2c, f, l; 3b, d-e, i; 4j-l; 7a-j; EV 2e, i; EV 3i; EV 4b-c, f, h; EV 5h.
 - Please indicate the statistical test used for data analysis in the legends of figures 2c, h, j, l; 3b, d-e, i; 4b, g, j-l; 5c-d; EV 2e, g, i; EV 3f-g, i; EV 4b-c, f, h; EV 5c-d, h-i.
 - Please note that in figures 2l; 7a-b; EV 5i; there is a mismatch between the annotated p values in the figure legend and the annotated p values in the figure file that should be corrected.
 - Please provide information on the nature and number of replicates in the legends of figures 3b, d-e; EV 4f.
 - Please describe the nature of replicates in the legends of figures 2h, j, l; 7c-j; EV 2e, g, i; EV 3f-g, i; EV 4b-d, h; EV 5i.
 - Please define the error bars in the legends of figures EV 1a, c; EV 4f.
 - Please define the scale bar for figure EV 2b.
 - Please define the white arrows in the legend of figure EV 1h.
11. Papers published in The EMBO Journal are accompanied online by a 'Synopsis' to enhance discoverability of the manuscript. It consists of A) a short (1-2 sentences) summary of the findings and their significance, B) 3-4 bullet points highlighting key results and C) a synopsis image that is 550x300-600 pixels large (width x height, jpeg or png format). You can either show a model or key data in the synopsis image. Please note that the image size is rather small and that text needs to be readable at the final size. Please send us this information together with the revised manuscript.

With best wishes,

leva

We realize that it is difficult to revise to a specific deadline. In the interest of protecting the conceptual advance provided by the work, we recommend a revision within 3 months (12th Oct 2024). Please discuss the revision progress ahead of this time with the editor if you require more time to complete the revisions.

Referee #1:

Now the revised manuscript satisfied some of my concerns on the data presentation. I appreciate the manuscript is data rich. Although technical points were improved, it is still not clear what is the direct consequence of loss of KDM2A in juvenile and adult testes. Demethylation of histone at H3K36 directly and indirectly affect the authors proposed that KDM2A, therefore leading to pleiotropic effects on several key steps in spermatogenic cell development. This reviewer does not doubt an importance of KDM2A in spermatogenic cell development. It would be more informative if the causal relationships on these defects should be more clearly distinguished. This reviewer suggests the title is not fully supported by the data and the main message in the title should be reconsidered.

Referee #3:

The authors responded to Reviewer 3 critique sufficiently.

Reviewer 1 pointed out some important considerations summarized as follows:

- The data presentation is not sufficient without appropriate clarifications.
- The transcriptome analysis may show population difference rather than bona fide DEG at the stage when the primary defect appeared.
- The direct effect of KDM2A on germ cell deficiency is still unclear. In response to these criticisms, the author provided additional supporting data and reorganized the manuscript.

In response, the authors tested the purity of c-KIT+ cells isolated from juvenile (P10) mouse testes, and the new results showed that the purity of c-KIT+ cells was over 90% in both the control and cKO groups, ruling out the possibility that the DEGs observed in the cKO group were caused by cell population differences. They also performed several additional sets of biological replicates on the phenotype analyses of Kdm2a cKO mice and confirmed that the differentiation of spermatogonia during the first round of spermatogenesis was not affected by KDM2A ablation and the primary defect they observed was the significant decrease in the number of preleptotene spermatocytes in the Kdm2a cKO mice, suggesting that meiotic initiation was severely affected by Kdm2a deletion.

The authors have satisfied these critiques.

POINT-BY-POINT RESPONSE TO REVIEWERS AND EDITORS

REVIEWER COMMENTS

Referee #1:

Now the revised manuscript satisfied some of my concerns on the data presentation. I appreciate the manuscript is data rich. Although technical points were improved, it is still not clear what is the direct consequence of loss of KDM2A in juvenile and adult testes. Demethylation of histone at H3K36 directly and indirectly affect the authors proposed that KDM2A, therefore leading to pleiotropic effects on several key steps in spermatogenic cell development. This reviewer does not doubt an importance of KDM2A in spermatogenic cell development. It would be more informative if the causal relationships on these defects should be more clearly distinguished. This reviewer suggests the title is not fully supported by the data and the main message in the title should be reconsidered.

Response: We apologize to the reviewer for not providing more information of the causal relationships on the defects we observed to negate their concerns but thank the reviewer for the valuable comments and suggestions to improve the quality of our manuscript. We would like to point out that our manuscript has 2 main conclusions 1) KDM2A loss results in dysregulation of some key genes essential for meiotic initiation (such as *Stra8* and *Meiosin*) and progression (such as *Sycp1* and *Spo11*), and 2) KDM2A ablation affects the deposition of H3K36me_{2/3} and the recruitment of the transcription factor E2F1 and the cofactor H2F1 to the promoters of these genes. We have carefully crafted out manuscript to not make claims beyond these conclusions. We feel that the addition of the information will not change the conclusions nor is it necessary for the claims made in the manuscript.

As suggested by this Reviewer, we have revised the Title of the manuscript to avoid overstatement, and we have also revised the Abstract and Discussion sections accordingly.

Referee #3:

The authors responded to Reviewer 3 critique sufficiently.

Reviewer 1 pointed out some important considerations summarized as follows:

- The data presentation is not sufficient without appropriate clarifications.
- The transcriptome analysis may show population difference rather than bona fide DEG at the stage when the primary defect appeared.
- The direct effect of KDM2A on germ cell deficiency is still unclear. In response to these criticisms, the author provided additional supporting data and reorganized the manuscript.

In response, the authors tested the purity of c-KIT⁺ cells isolated from juvenile (P10) mouse testes, and the new results showed that the purity of c-KIT⁺ cells was over 90% in both the control and cKO groups, ruling out the possibility that the DEGs observed in the cKO group were caused by cell population differences. They also performed several additional sets of biological replicates on the phenotype analyses of Kdm2a cKO mice and confirmed that the differentiation of spermatogonia during

the first round of spermatogenesis was not affected by KDM2A ablation and the primary defect they observed was the significant decrease in the number of preleptotene spermatocytes in the Kdm2a cKO mice, suggesting that meiotic initiation was severely affected by Kdm2a deletion.

The authors have satisfied these critiques.

Response: We thank Reviewer #3 for his original comments and for accepting our responses. The manuscript is much improved now that all the Reviewers' suggestions have been incorporated.

Dear Dr. Yuan,

Thank you for addressing the final editorial issues. I am now pleased to inform you that your manuscript has been accepted for publication.

Before we forward your manuscript to our publishers, I would like to propose some minor edits in the manuscript title, abstract and synopsis (please see below and the attached manuscript text file). I have also written a short blurb that will accompany the title of your manuscript in our online system. Please let me know if any corrections or adjustments are needed.

Title:

Histone demethylase KDM2A recruits HCFC1 and E2F1 to orchestrate male germ cell meiotic entry and progression

Blurb:

The balance of H3K36 di- and tri-methylation regulates meiotic gene expression during mouse spermatogenesis.

Synopsis:

The role of histone demethylation in spermatogenesis remains incompletely understood. This work shows that KDM2A demethylates trimethylated histone H3 lysine 36 (H3K36me3) and cooperates with the transcription factor E2F1 during the transition from mitosis to meiosis.

- KDM2A is essential for meiotic initiation and progression by promoting meiotic gene expression and suppressing the expression of genes involved in spermatogonial development.

- In male germ cells, KDM2A specifically demethylates H3K36me3, thus increasing H3K36me2 levels.

- KDM2A interacts with E2F1 and its co-factor HCFC1 to facilitate meiotic gene expression and ensure proper meiotic progression.

Finally, we would like to promote your manuscript among the Chinese readership. Therefore, we would like to invite you to prepare a short summary of the manuscript in Chinese (1500-2000 Chinese characters), which we will promote on the WeChat platform 'BioArt' with more than 610,000 followers.

If you are interested in this opportunity, we recommend covering the article very close to its online publication date. Thus, ideally we would very much appreciate if you could send us a draft within the next 7 working days. Please let us know whether or not you would be interested in contributing such a short summary in Chinese.

I have included below some general guidelines on how to prepare a summary and a link to recent examples for your reference. Please let me know if you have any questions about this.

If you have any questions, please do not hesitate to contact the Editorial Office. Thank you for this contribution to The EMBO Journal and congratulations on a nice study!

Best wishes,

Ieva

Ieva Gailite, PhD
Senior Scientific Editor

The EMBO Journal
Meyerhofstrasse 1
D-69117 Heidelberg
Tel: +4962218891309
i.gailite@embojournal.org

General WeChat Summary Guidelines

1. These summary articles are meant to be targeting general audience so please limit the use of specialized technical terms, acronyms and jargon.
2. A summary usually starts with brief background information of the reported work, which is followed by explaining the findings in some detail, and ends with a short review of the conclusions as well as the implications of the work and future directions for the research.
3. The summary should at least contain one graphical item, such as a scheme or a figure from the paper.
4. Please provide ONE SINGLE document containing all text and graphical materials, ideally as a Word.docx or .doc file. Please DO NOT provide the document as a .pdf file.
5. Please DO NOT publicly release the document before the paper is officially published online.

Summary Examples

EMBO J | 罗招庆/欧阳松应揭示谷酰胺脱氨酶MvcA的去泛素化功能

EMBO J | 王松灵院士团队揭示组织内应力调控大型哺乳动物乳恒牙替换的新机制
